# The cryptic seismic potential of the Pichilemu blind fault in Chile revealed by off-fault geomorphology

J. Jara-Muñoz [1,7 ✉], D. Melnick [2], S. Li [3], A. Socquet [4], J. Cortés-Aranda [5], D. Brill [6] & M. R. Strecker[1]

The first step towards assessing hazards in seismically active regions involves mapping capable faults and estimating their recurrence times. While the mapping of active faults is commonly based on distinct geologic and geomorphic features evident at the surface, mapping blind seismogenic faults is complicated by the absence of on-fault diagnostic features. Here we investigated the Pichilemu Fault in coastal Chile, unknown until it generated a Mw 7.0 earthquake in 2010. The lack of evident surface faulting suggests activity along a partly-hidden blind fault. We used off-fault deformed marine terraces to estimate a fault-slip rate of $0.52 \pm 0.04$ m/ka, which, when integrated with satellite geodesy suggests a $2.12 \pm 0.2$ ka recurrence time for Mw~7.0 normal-faulting earthquakes. We propose that extension in the Pichilemu region is associated with stress changes during megathrust earthquakes and accommodated by sporadic slip during upper-plate earthquakes, which has implications for assessing the seismic potential of cryptic faults along convergent margins and elsewhere.

[1] Department of Earth Sciences, University of Potsdam, Karl-Liebknecht-Str. 24-25, Potsdam, Germany. [2] Austral University of Chile, Institute of Earth Sciences, Edificio Pugín, Campus Isla Teja, Valdivia, Chile. [3] State Key Laboratory of Lithospheric Evolution, Institute of Geology and Geophysics, Chinese Academy of Sciences, No. 19, Beitucheng Western Road, Beijing, China. [4] University of Grenoble Alpes, University Savoie Mont Blanc, CNRS, IRD, UGE, ISTerre, 38000, Grenoble, France. [5] Department of Earth Sciences, Universidad de Concepción, Víctor Lamas 1290, Concepcion, Chile. [6] Institute of Geography, University of Cologne, Otto-Fischer-Straße 4, Cologne, Germany. [7] Present address: Department of Civil Engineer, Hochschule Biberach, Karlstraße 9-11, Biberach, Germany. ✉email: jara@geo.uni-potsdam.de

Unexpected ruptures that occur during large-magnitude earthquakes along previously unmapped faults emphasise a major lacuna in our knowledge concerning the location and seismic potential of tectonically active structures[1]. Over the past decade, earthquake ruptures have occurred along previously unidentified or not fully mapped faults during at least five Mw > 6 earthquakes. These include the 2010 Pichilemu earthquakes in Chile, the 2016 Kaikoura earthquake that ruptured the unmapped Papatea Fault, the 2010 Darfield and 2011 Christchurch earthquake sequence in New Zealand, and the 2019 Ridgecrest events in California[2–6]. These earthquakes highlight an important gap in our understanding of the seismogenesis of hidden faults in a variety of geodynamic environments. Whereas mapping potentially active faults commonly relies on identifying geomorphic and geologic features indicative of surface rupture and deformation that could be associated with past earthquakes[7,8], in areas with active blind faults such evidence may either be completely absent or difficult to identify[9,10]. Blind faults are geological structures whose ruptures do not reach the earth's surface[11], thereby hiding their seismogenic potential. Such structures are common in sedimentary basins and have often been identified on the basis of geophysical imagery[12–14] and indirect geomorphic observations[15]. However, estimating the seismic potential of blind faults is difficult in the absence of any on-fault geologic or geomorphic evidence from past earthquakes. Here we demonstrate that quantifying deformation using off-fault geomorphic strain markers provides valuable insight into cryptic, potentially active faults in coastal areas. We use the terms on-fault and off-fault to distinguish between surface deformation that has occurred along the fault trace and within the zone surrounding the fault, respectively.

The polarity and spatial distribution of stresses in the upper plate of subduction zones changes throughout the seismic cycle[16]. For instance, in response to the nearly instantaneous polarity change associated with a megathrust earthquake, the upper plate is commonly affected by enhanced extension that results in increased seismicity and the triggering of occasional crustal earthquakes[17,18]. Slip on upper-plate faults triggered by megathrust earthquakes has been reported in Japan, Alaska, and Chile[18–20], and has been inferred to be a common feature along most subduction zones[18,21–23]. However, historical and paleoseismic observations suggest that crustal faults with a low slip rate are characterised by recurrence times involving thousands of years, and may therefore not be reactivated during every megathrust earthquake, since these commonly recur over periods ranging from a number of decades to a few centuries[18,22,24]. Because both exposed and hidden upper-plate faults are widespread along coastlines bordering subduction zones, they pose significant local hazards. Such crustal faults can produce higher amplitude seismic waves at local scales than megathrust earthquakes and may locally increase the amplitude and shorten the arrival times of tsunamis in the near-field[25]. Mapping crustal structures along subduction zones and quantifying their slip rates and relationships to megathrust earthquake cycles is therefore a fundamental requirement for obtaining an adequate assessment of the spatiotemporal characteristic of earthquake and tsunami hazards.

In this study we focus on the Pichilemu Fault (PIF), a hidden fault that was unknown until it generated two shallow Mw 7 and 6.9 earthquakes 11 days after the Mw 8.8 Maule megathrust event that affected central Chile in 2010. Our study combines geomorphic and morphometric analyses using high-resolution LiDAR topography, luminescence dating, radar interferometry, and numerical modelling. We show that while on-fault displaced geomorphic markers are absent along the surface fault traces, off-fault strain markers can be used to estimate a long-term slip rate,

which, when integrated over the 2010 coseismic deformation pattern and assuming a characteristic slip behaviour, allows a recurrence rate to be inferred for such earthquakes. Our results demonstrate the hidden seismogenic potential of blind faults, with implications for seismic hazard along coastlines bordering subduction zones.

## Results and discussion
**Seismotectonic and geologic setting.** The PIF is located at the coast of the central Chile margin, where the Nazca plate is subducting beneath South America at 66 mm/yr[26] (Fig. 1A). This region comprises the Coastal Range, which reaches maximum elevations of ~600 metres above sea level (MASL) and consists almost exclusively of crystalline metamorphic rocks and scattered intrusive bodies; the metamorphic rocks are related to a Paleozoic accretionary prism overprinted by brittle deformation during Mesozoic and Cenozoic exhumation[27–29] (Fig. S1A). The seaward slope of the range is sculpted by a sequence of uplifted marine terraces, some of them overlain by shallow marine deposits[30]. Dense vegetation and thick soil cover have hampered the mapping of geological and geomorphic features in the area, resulting in different interpretations regarding the presence of tectonically active structures e.g., refs. [4,30].

The Pichilemu area was affected by the 2010 Maule earthquake (Mw 8.8), which ruptured a ~500 km-long portion of the megathrust, with a peak slip of 17 m to the south of the PIF[31] (Fig. 1A). The Maule earthquake triggered instantaneous slip along the Santa Maria fault and non-instantaneous slip along the PIF, which are located in the southern and northern parts of the rupture zone, respectively[4,19] (Fig. 1A). Prior to the 2010 earthquake, only a few faults affecting the crystalline basement and Cenozoic sedimentary cover were known in this area[28,32] (Fig. 1B); the PIF was unknown.

Eleven days after the Maule earthquake, the PIF slipped during two Mw 6.9 and 7 normal-faulting earthquakes (Fig. 1B), followed by ~12,000 aftershocks located between the megathrust and a depth of ~4 km[33] (Fig. 1C). The aftershocks delineated a NW-striking, SW-dipping structure extending for ~80 km along strike. Surface displacements estimated from GPS and ALOS/PALSAR Interferometric Synthetic Aperture Radar (InSAR) collected 2 days before and 44 days after the Maule earthquake suggest that the 2010 PIF earthquakes were associated with a maximum slip of ~3 m along the main strand of the PIF, extending from 5 to 22 km depth[34]. However, despite the large magnitude of the PIF earthquakes, no evidence of surface ruptures was found during field surveys nor were any detected in radar interferometry images[4,34], which suggests that the PIF is a blind, yet tectonically active structure.

**On-fault tectonic geomorphology.** We used a Digital Terrain Model (DTM) derived from airborne Light Detection And Ranging (LiDAR) data at 1 m resolution to estimate fluvial metrics[35] and analyse the surface expression of the PIF (see Methods Section, Analysis of on-fault geomorphic features, and Figs. S1A–F and S2A–E). The coastal reaches of the PIF comprise two catchments of ~100 km² each (C1 and C2; Fig. 2A) developed almost exclusively on metamorphic bedrock, except for the western part of C2, which contains isolated intrusive outcrops, and near the coast, where the valley floors are filled by marine, fluvial, and aeolian deposits (Fig. S1A). The median annual rainfall is similar in both catchments, with values of 622 mm/yr in C1 and 618 mm/yr in C2, based on data from the Tropical Rainfall Measuring Mission (TRMM)[36] (Fig. S2E). The catchment asymmetry is evident in each of these catchments and manifested in the deviation between the main trunk stream and the

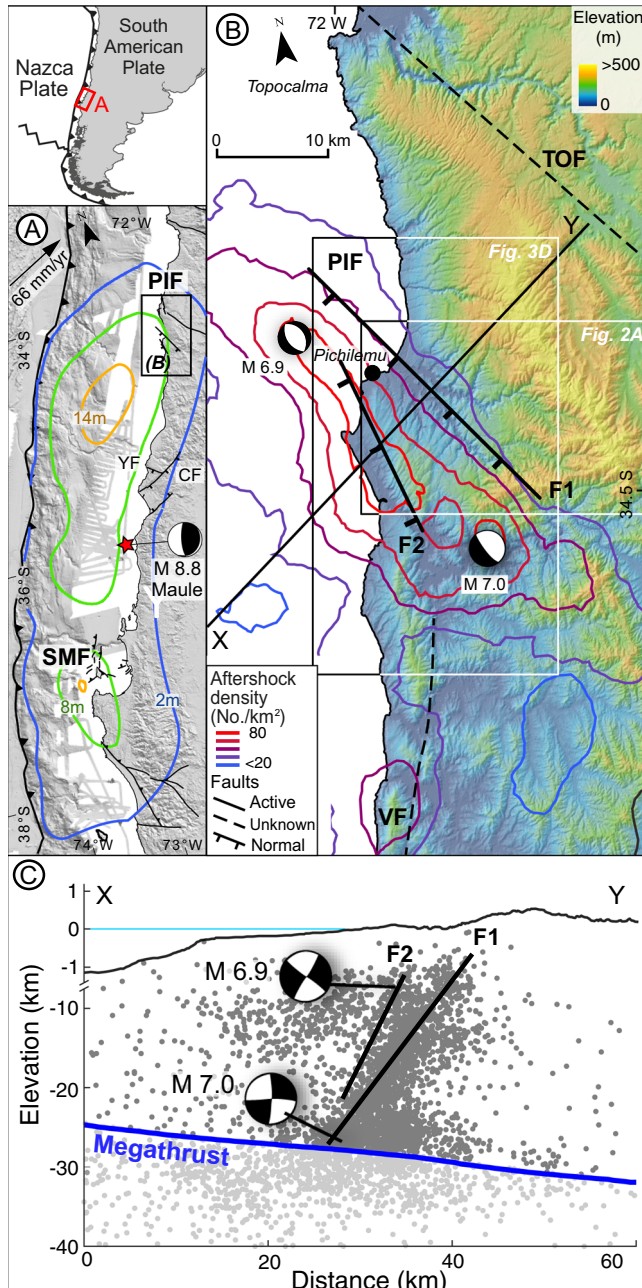

**Fig. 1 Tectonic setting of the Pichilemu Fault. A** Slip distribution for the 2010 Maule earthquake[31] and active faults from the Chilean Active Fault database[58]. The Santa María Fault (SMF) and Pichilemu Fault (PIF) were reactivated by the earthquake[4,19]. YF: El Yolki Fault; CF: Carranza Fault. **B** Topography (SRTM data from www2.jpl.nasa.gov), faults, and focal mechanisms of the PIF earthquakes[4] in the Pichilemu region. Contours show aftershock density after the Maule earthquake. Thick black lines denote the PIF branches (F1 and F2). Also shown are the Topocalma Fault (TOF), a Quaternary normal fault, and Vichuquén Fault (VF), an inferred fault affecting the crystalline basement[4]. **C** Crustal profile showing aftershocks (Mw > 1) recorded between March 15 and September 30, 2010[33], used to infer the subsurface geometry of the PIF. Megathrust from Slab 2.0 model[47].

catchment centreline, which is quantified using the symmetry factor (Ts)[37] (see Methods Section, Analysis of on-fault geomorphic features). The asymmetry of C1 and C2 is highlighted by trunk streams that converge along a section parallel to the trace of the PIF (Ts > 0.6); further east, the catchments become

progressively symmetric with Ts~0.1 (Fig. 2A and Fig. S2B). The area to the west of the centreline in C2 includes NE-SW-elongated parallel drainages that are probably associated with local surface tilting. Catchment C1 is characterised by higher local relief than C2 with median values of 111 and 135 m, respectively, and reaching 300 m in C1 and 250 m in C2 (Fig. S2C). The median slope of both catchments is similar (13° for C1 and 14° for C2), but its distribution is slightly biased towards lower values in catchment C2 (Fig. S2D). The drainage network of catchment C1 reaches ~500 MASL and includes 16 knickpoints distributed between 100 and 350 MASL (Fig. 2C). Steepness index ($K_{sn}$) values reach up to 100, with the higher values forming a fringe at elevations between 300 and 400 MASL. C2, in contrast, includes eight knickpoints distributed between 100 and 150 MASL and has $K_{sn}$ values of up to 70 (Fig. 2A). A conspicuous set of NW-SE and NE-SW-oriented lineaments can be identified from aligned drainages and small slope breaks. The westernmost lineaments (L1 and L2; Fig. 2A and Fig. S1), are ~3 km-long and have ~50 m-high scarps partly degraded by river incision and associated with contacts between metamorphic and intrusive rock units (Fig. S1A). Their traces are oblique to fault geometries inferred from crustal seismicity (F1 and F2). Farther east, lineaments L3, L4 and L5 are highlighted by aligned valleys and a trellis drainage pattern (Fig. 2A, E and Fig. S1). These lineaments extend for ~7 km along strike, are associated exclusively with metamorphic rocks, and are subparallel to local metamorphic foliations[27,38] (Fig. S1A), but oblique to the trace of F1.

Catchment and drainage metrics indicate variable degrees of surface deformation in this area. For instance, the high symmetry factors of both catchments together with the subparallel drainages in the western part of C2 suggest local tilting of the PIF footwall and hanging-wall blocks in opposite directions (Fig. 2 and Fig. S2B). The difference in local relief and catchment slope between C1 and C2 reflect the varying degrees of river incision, which, together with the differences in $K_{sn}$ values, knickpoint locations, and drainage elevations, may indicate differential vertical displacements. The minimum differences between the catchments in terms of rainfall and metamorphic-bedrock lithology (100% of the area in C1 and ~75% in C2; Fig. S1A) suggest that these two factors exert a negligible control on catchment asymmetry and drainage metrics.

A detailed analysis of the PIF lineaments suggests that these features cannot be directly interpreted as fault scarps associated with surface ruptures during recent earthquakes. The drainages crossing lineaments L1 and L2 are associated with low $K_{sn}$ values, suggesting that if these lineaments were related to active faults, either the faults are slipping at low rates that have a little effect on river incision, or they are associated with a blind fault that does not reach the surface. Furthermore, the lineaments coincide with lithological contacts, indicating that their morphology might respond to differential erosion rather than the effects of surface faulting. Considering their orientations and morphologies, lineaments L3 to L5 are the best candidates for fault scarps. However, even though they are associated with subtle breaks in slope, a well-developed scarp cannot be distinguished on scarp-perpendicular swath or ridge-crest profiles (Fig. 2E, F). Furthermore, no clear spatial relationship exists between the trace of these lineaments and the fringe of higher $K_{sn}$ values of C1 regarding overlap or orientation (Fig. 2A).

Chi-plots are commonly used to identify transient signals propagating upstream along fluvial systems, such as tectonically generated knickpoints[39] (see Methods Section, Analysis of on-fault geomorphic features for further details). However, we found no relationship between chi-plot ($\chi$) values at the knickpoint locations in C1 and the distance from the knickpoints to the mapped lineaments (Fig. 2B and Fig. S2A). Furthermore, the

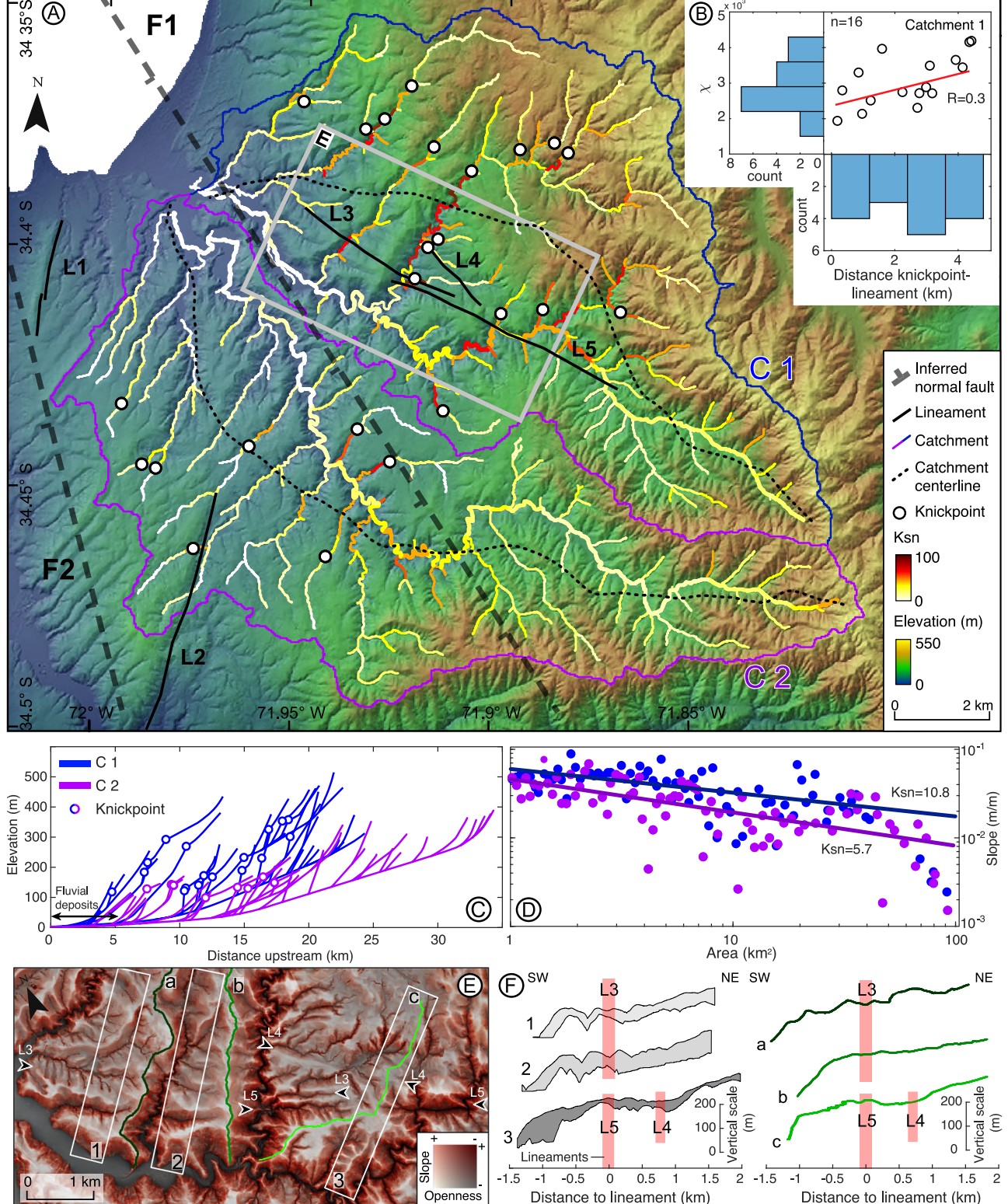

**Fig. 2 On-fault tectonic geomorphology from bare-earth LiDAR topography. A** Shaded-relief map of the PIF coastal reach showing the two analysed catchments (C1 and C2), the thick dashed black lines are the faults (F1 and F2) inferred from spatial alignments of crustal seismicity. Note asymmetry of the main trunk streams with respect to the catchment centreline. The black lines labelled L1 to L5 denote lineaments that may represent surface-breaching faults (see text for discussion). Steepness index ($K_{sn}$) determined for a reference concavity of 0.45. **B** Scatter plots and histograms comparing fluvial metrics with the lineament metrics within C1. Scatter plot and histograms of chi-values ($\chi$) at knickpoints with respect to distance to lineament (See chi-map Fig. S2A). **C** Longitudinal river profiles of both catchments showing knickpoint locations. **D** Slope-area plots for both catchments. Note that C1 has a higher $K_{sn}$ value suggesting more rapid uplift. **E** Red-relief map of lineaments, ridge profiles (green lines) and swath boxes (black rectangles) shown in **F**. The arrows indicate the trace of the lineaments. **F** Swath and ridge-crest profiles. Profiles 1, 2, a, and b are centred on the trace of lineament L3, and profiles 3 and c are centred on the trace of L5. Note the lack of evident scarps suggesting an absence of any recent surface-breaching ruptures.

linear distances between knickpoints and the L3 and L5 lineaments do not display a clear trend (Fig. 2B), suggesting that knickpoints are not related to the potential active fault scarps but rather reflect the effects of base-level changes associated with relative sea-level variations. The results of our morphometric analyses of LiDAR topography indicate that the area has been affected by surface deformation with a degree of spatial asymmetry, which may be a result of tilting and differential uplift. However, these deformation patterns cannot be directly related to any particular structure with a marked surface expression. We therefore conclude that this area is not characterised by localised deformation at the surface, but rather by strain distributed over a 10 km-wide region.

**Off-fault tectonic geomorphology.** Uplifted marine terraces—geomorphic markers of past relative sea-level positions[40]—are ubiquitous along the coast of central Chile[30] and can be used as regionally correlatable strain markers[41]. We mapped terraces at Pichilemu using a LiDAR DTM and the TerraceM-2 software[42] (Fig. 3A, see Methods Section, Analysis of off-fault geomorphic features). To the south of the PIF there are four levels of wave-cut terraces sculpted into the bedrock reaching up to 100 MASL (Fig. 3B and Fig. S3); these are occasionally covered by a thin veneer of marine and aeolian sediments. In contrast, to the north of the PIF six distinct sedimentary units corresponding to wave-built terraces are exposed; they consist of shallow marine sandstone bodies onlapping against the crystalline bedrock at elevations between 50 and 170 MASL, each comprising a single regressive cycle (Fig. 3B, Fig. S4A–C). From the lower and intermediate terrace levels at 51 and 115 MASL, respectively, we obtained post-IR IRSL (post-infrared infrared stimulated luminescence, see Methods Section, Post-IR IRSL dating) ages of 106 ± 9.3 and 297 ± 29 ka, which correspond to Marine Isotope Stages (MIS) 5d and 8 (Fig. 3A, Figs. S4A, S5, and S6, Table 1). These sediments were deposited above bedrock during shoreline progradation shortly after the corresponding MIS highstand e.g., refs. [30,43]. We have correlated the surface morphology and geometry of these deposits with MIS 5e and MIS 9, at 125 and 320 ka respectively (Fig. 3A, B, Figs. S4, and S6). We have also tentatively correlated two additional terrace levels with MIS 7 and MIS 11 based on a composite sea-level curve (Fig. S6), these relative sea-level highstands correspond to ages between 250 and 380 ka.

South of the PIF, the lower marine terrace level is continuously exposed between Punta de Lobos and La Puntilla, with widths ranging between 1 and 3 km. This surface has been previously interpreted as a rasa[30], i.e., a terrace surface formed by marine reoccupation during successive highstands. Marine sediments that cover this rasa level between 16 and 32 MASL have yielded post-IR IRSL ages of 328 ± 33 and 314 ± 30 ka, corresponding to MIS 9 (Fig. 3A, Figs. S5 and S6). The upper terrace levels are characterised by well-defined paleo-cliffs and narrow paleo-platforms, with mean shoreline-angle elevations of 42, 60, and 80 MASL decreasing southward. By correlating the sequence with global sea-level curves[44,45], we interpret the age of the three upper terrace levels to range between MIS 11 and MIS 19 (Fig. S6).

The estimated uplift rates vary between 0.06–0.15 m/ka and 0.37–0.46 m/ka across the PIF (Fig. 3C), with associated 2 s errors between 0.01 and 0.08 m/ka, suggesting protracted emergence over the past ~620 ka. Overall, the marine terrace sequence displays a broad warping pattern with a wavelength of ~10 km, which is compatible with rapid uplift and back tilting along the PIF footwall block, and monoclinal rollover folding in the hanging wall, which is consistent with a NW-striking, SW-dipping normal fault at depth (Fig. 3B, C).

**Coseismic slip and long-term slip rate of the PIF.** Using a combination of coseismic displacements derived from GPS and InSAR together with fault geometries inferred from aftershock seismicity[33,46], we estimated slip along the PIF during the 2010 earthquakes (Figs. 3D, 4A–B, Figs. S7A–E and S8A–F). The InSAR data comprised two Envisat® scenes acquired 2 days before and 7 days after the PIF earthquakes, obtaining an ascending interferogram (See Methods Section, Estimating coseismic slip during the PIF earthquakes). The aftershocks clearly mark the down-dip termination of the PIF at ~26 km depth where it intersects the megathrust[47]; in contrast, the up-dip limit is more diffuse and most likely located at a depth between 5 and 8 km (Fig. S7B–E). We carried out a set of forward models by iterating up-dip depths and slip magnitudes for F1 and F2 within pre-defined ranges (see Table S1 and Methods Section, Estimating coseismic slip during the PIF earthquakes). We varied the along-dip extent of the fault by moving the up-dip limit up and down along fixed dips (55° for F1 and 72° for F2), strikes (N38°W for F1 and N16°W for F2), and down-dip depths (26.2 km for F1 and 20 km for F2). The model allowed us to estimate a slip magnitude and up-dip depth for each fault, along with their corresponding uncertainties, using the normalised root mean squared error (NRMSE, Figs. S9A and S10A–D). The model results suggest that F1 slipped 1.1 m (with 90% confidence interval (CI) between 0.95 and 1.15 m) between the megathrust and a depth of 4.6 km (CI: 3.6–5 km, Figs. S9A and S10B); and F2 with 0.1 m of slip (CI: 0–0.4 m) and 5.2 km up-dip (CI: 2.2–5.4 km). The 1.1 m coseismic slip of F1 obtained by forward modelling is similar to the median slip of 1.2 m predicted by the inverse model (Fig. S11A–F).

In order to evaluate a possible dependency between the slip of the F1 and F2 faults and to rule out a trade-off, we tested their fault-slip variability as independent variables with respect to the NRMSE. We observed variabilities of ~400% and ~10% for F1 and F2 as independent variables, respectively, suggesting a strong dependency of F2 with respect to slip values of F1 lower than 1.1 m (Fig. S12). However, our best-fit model suggests that the slip magnitude of F2 is one order of magnitude smaller than that of F1, and therefore we conclude that the F2 trade-off has a minimal effect on our results.

The forward model residuals are <0.05 m in the near-field (within 10 km of the fault), increasing to ~0.1 m in the far-field. (Fig. S8F). The higher far-field residuals are probably associated with post-seismic deformation following the 2010 Maule earthquake, which was not considered in our model. However, because of its deep source (at the megathrust and continental mantle[48]), post-seismic deformation is associated with wavelengths of ~100 km[48,49], which is an order of magnitude greater than deformation related to the shallower PIF (Fig. 4A, B). We therefore consider that our model adequately reproduces the near-field surface deformation resulting from coseismic slip along the PIF. Our model results suggest that the PIF earthquakes occurred along blind faults, whose deformation pattern differ from those associated with surface-breaching faults (see comparison in Fig. S13A, B).

We estimated a long-term slip rate for the PIF (over the past ~500 ka) by forward modelling the spatial pattern of uplift rates estimated from marine terraces. We modelled surface deformation associated with the same two faults used in the coseismic model (F1 and F2; Fig. 4C, D), which clearly offset the elevations of marine terraces. We generated two models, one based on uplift rates estimated from measured shoreline angles, and a second one, based on an interpolated surface derived from these measurements to increase the area available for comparing measurements with model results (Fig. 4D, see Methods Section, Analysis of off-fault geomorphic features). In the first model, we

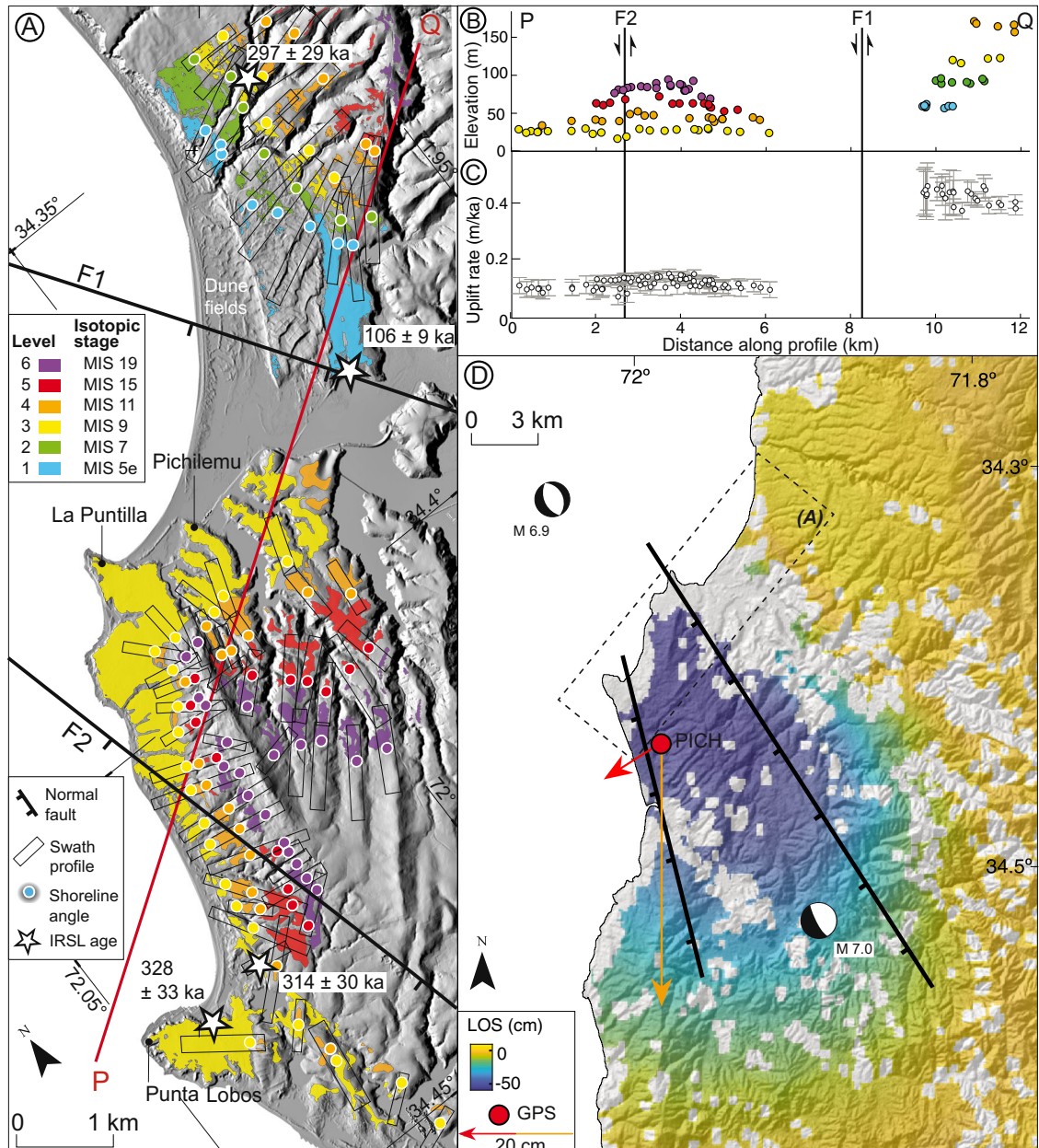

**Fig. 3 Coseismic and long-term surface deformation associated with the PIF. A** Map of the six levels of marine terraces in the PIF area and ages assigned to Marine Isotope Stages (MIS). Boxes show the location of swath profiles used to map shoreline angles (colour-coded points). Thick black lines denote the two PIF branches (F1 and F2). **B** Shoreline angles projected along profile P–Q; arrows indicate relative displacements of F1 and F2. **C** Uplift rates deduced from shoreline angles along profile P–Q, grey lines are standard error estimates of uplift rates (se calculation details in Methods Section Analysis of off-fault geomorphic features). **D** Coseismic Line-of-Sight (LOS) displacements in the Pichilemu area determined from Envisat® radar interferometry of the 2010 PIF earthquakes. Red and orange arrows represent horizontal and vertical displacements, respectively, estimated at the continuous GPS station PICH (red dot) as a result of the 2010 PIF earthquakes[34].

obtained a slip rate of 0.48 m/ka for F1 (CI: 0.46–0.54 m/ka), for a fault extending from 26.2 to 1.2 km depth (CI: 0.8–2.6 km) and 0.2 m/ka for F2 (CI: 0.08–0.22 m/ka) with an up-dip depth of 3.6 km (CI: 1.2–3.8 km). The second model yielded a slip rate of 0.52 m/ka (CI: 0.48–0.56 m/ka) for F1 and 0.1 m/ka for F2 (CI: 0.08–0.2 m/ka; Figs. S9B and S10E–H). The best-fit up-dip slip depths were 1.8 km for F1 (CI: 1–2.6 km) and 1.2 km for F2 (CI: 0.8–3.4 km F1; Figs. S9C and S10I–L). Importantly, both models predict similar deformation patterns and slip rates, and these differed from those predicted for a surface-breaching fault (Figs. S13C, D). We selected the second model as the best-fit that optimally reproduces the deformation pattern of marine

terraces, with lower uncertainties than the first model and convergence towards low NRMSE values (Fig. 4D and Fig. S9C).

Marine terraces in the Maule region record two wavelengths of deformation; long wavelength deformation patterns (>100 km) associated with deformation along the coast controlled by the megathrust, and short-wavelength patterns associated with crustal faults[30]. The pattern of deformed marine terraces in the PIF area exhibits a short deformation wavelength of ~10 km, which is one order of magnitude smaller than the deformation related to the megathrust. The effect of megathrust deformation is therefore negligible compared to the shorter wavelength of crustal faults.

**Table 1 Post-IR IRSL samples.**

| Sample | Long (deg) | Lat (deg) | Z (m) | Depth (m) | Paleodose (Gy) | No. aliquots | Over dispersion (%) | U (ppm) | Th (ppm) | K (%) | Water (%) | Dose rate (Gy/ka) | Age (ka) |
|---|---|---|---|---|---|---|---|---|---|---|---|---|---|
| PICH-1 | −71.97 | −34.38 | 51 | 2.80 | 242 ± 13 | 5 | 2 | 1.44 ± 0.07 | 4.94 ± 0.28 | 0.86 ± 0.03 | 5 ± 5 | 2.3 ± 0.1 | 106 ± 9.3 |
| LOBO-1 | −72.04 | −34.42 | 14 | 3.20 | 748 ± 58 | 14 | 22 | 0.96 ± 0.05 | 3.61 ± 0.22 | 1.09 ± 0.04 | 12 ± 5 | 2.3 ± 0.1 | 328 ± 33 |
| PICH-2 | −71.96 | −34.33 | 115 | 2.20 | 706 ± 51 | 19 | 22 | 1.13 ± 0.07 | 5.32 ± 0.34 | 1.05 ± 0.01 | 6 ± 5 | 2.4 ± 0.1 | 297 ± 29 |
| PICH-4 | −72.03 | −34.42 | 38 | 6.5 | 784 ± 54 | 17 | 18 | 1.18 ± 0.07 | 5.28 ± 0.29 | 1.20 ± 0.02 | 9 ± 5 | 2.5 ± 0.1 | 314 ± 30 |
| SM mb | −73.51 | −37.04 | 2 | 0.5 | 3.4 ± 0.1 | 15 | 9 | – | – | – | – | – | – |

Radionuclide analyses include Uranium (U), Thorium (Th) and Potassium (K). Sample SM mb is a present-day beach berm sample from Santa Maria Island used to evaluate the completeness of signal resetting.

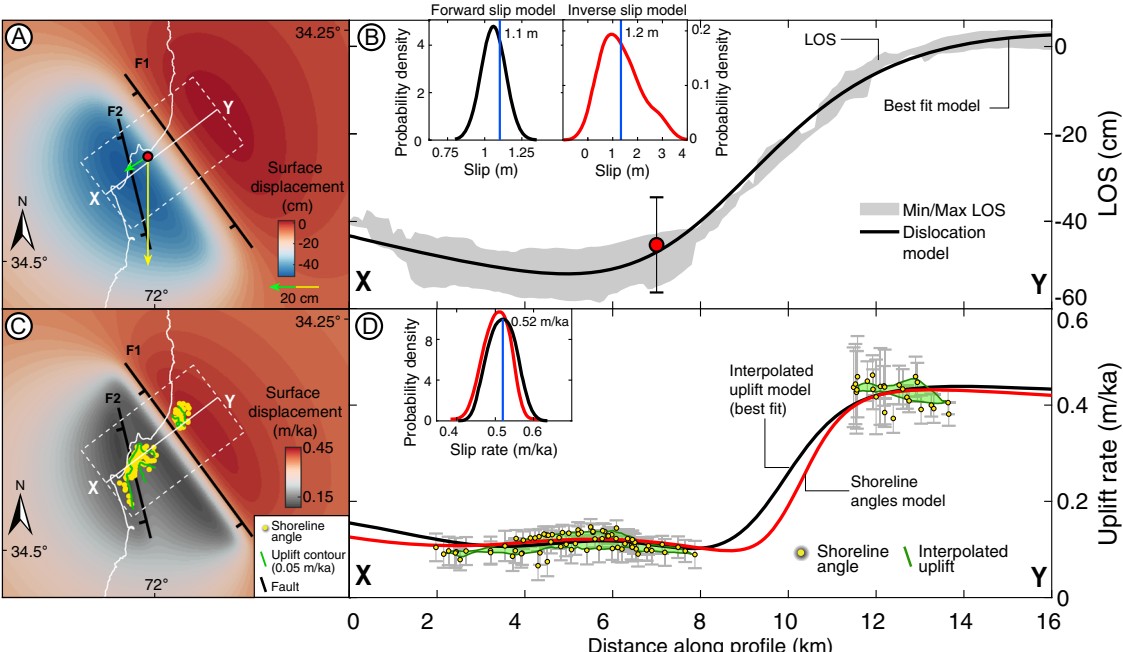

**Fig. 4 Coseismic and long-term fault-slip models. A** Best-fit model forward of Line of Sight (LOS) displacements as a result of the 2010 PIF earthquakes, green and yellow arrows denote horizontal and vertical displacements, respectively, estimated at continuous GPS station PICH[34]. **B** Swath profile showing Envisat® LOS displacements and best-fit forward model. Red dot shows PICH displacement projected to Envisat LOS. Insets in B are the probability-density distributions of coseismic slip of the forward (black line) and inverse (red line) models (See Figs. S10 and S11), blue lines denote best-fit slip and median slip of both models, respectively. **C** Best-fit dislocation model based on interpolated marine terrace uplift rates (yellow dots denote measurement locations). **D** Swath profile comparing fault-slip model based on shoreline angles (red line) with a model based on interpolated uplift rates from shoreline angles (black line). Insets show probability-density distributions of slip rate values used to estimate confidence interval (See Fig. S10). The green polygons in **D** are the interpolated uplift rate surfaces projected along the profile.

Our coseismic and long-term slip models both suggest blind faulting with similar surface deformation patterns. The differences in up-dip depths between both models may be associated with a partial rupture during the PIF earthquakes, as has been observed elsewhere in other crustal earthquakes e.g., refs. [50,51], or to the relatively simple model setups that assume heterogeneous slip along a planar fault with homogeneous rheology. The inverse model predicts a median slip of 1.2 m arranged into an irregular patch of slip centred at ~12.5 km depth that extends upwards to ~6 km depth (Fig. S11D). This is consistent with blind faulting, which is also indicated by both the coseismic and long-term forward models. Taken together, our models closely reproduce the pattern and magnitude of observed surface displacements, suggesting that deformation associated with the PIF occurs over a ~10 km-wide area.

**Earthquake recurrence of the Pichilemu fault.** Both the coseismic and long-term surface deformation patterns of the PIF are similar, and both are consistent with extensional fault

kinematics. We therefore propose that the PIF accrues permanent deformation only during slip triggered by megathrust earthquakes, such as during the events observed after the Maule earthquake. Historical, paleoseismic, and paleo-tsunami records suggest a recurrence time of ~0.1–0.2 ka for Maule-type events[52–54]. If we consider a recurrence time of 0.2 ka, then a triggered slip of 0.1 m would account for the long-term PIF slip rate (Fig. 5A). The inferred slip would be equivalent to offsets during a Mw 5.3–6.1 earthquake based on empirical relationships[55–57] (i.e., a seismic moment between $1.1 \times 10^{17}$ and $1.8 \times 10^{18}$ Nm, Fig. 5B). In turn, a recurrence time of 0.5 ka would imply 0.26 m of PIF slip (Fig. 5A) and an earthquake magnitude between Mw 5.9 and Mw 6.3 (Fig. 5B). In both cases the estimated slip per event would be five to ten times less than during the 2010 PIF earthquakes. Instead, the amount of slip per event required to trigger a Mw 7 earthquake would be 1.65 m; in this case, the recurrence time required to account for the long-term slip rate would be 3.2 ka. By combining the probability distribution of the long-term PIF slip rate with that of the coseismic slip we infer a recurrence time of 2.12 ka (1.85–2.28 ka at 90%

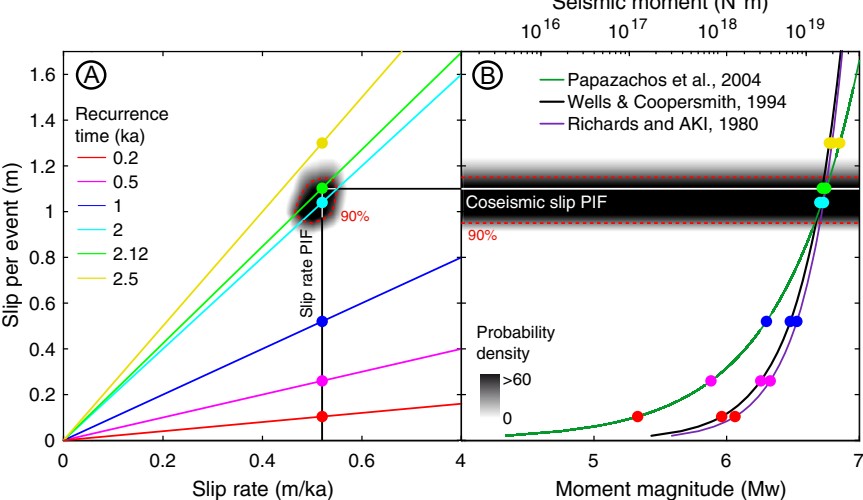

**Fig. 5 Recurrence time for earthquakes on the PIF. A** Recurrence times estimated for different values of coseismic slip values and long-term slip rates, assuming that PIF slip occurs only during megathrust earthquakes. **B** Equivalent seismic moment for each recurrence time in **A**, indicated by colour-coded points. Lines denote different empirical relationships (see references in text). The black-to-grey areas in **A**, **B** represent the probability-density of the PIF slip. The dashed red lines in **A**, **B** show the 90% confidence interval. The black line in **A** and white line (**B**) show the best-fit slip per event and slip rate, respectively.

confidence interval, Fig. 5A) for Mw 6.7 to Mw 6.8 earthquakes (seismic moment between $1.4 \times 10^{19}$ and $2.0 \times 10^{19}$ Nm; Fig. 5B), similar to that of the 2010 PIF earthquakes. Our results suggest that the recurrence time of the PIF may be over an order of magnitude greater than that of Mw > 8 megathrust earthquakes in the Maule segment, implying sporadic triggering of fault slip.

The forearc along the 2010 Maule rupture zone includes ten normal faults similar to the PIF[58] and associated with similar throws[30], slip rates[24,59] and lengths of their surface traces. If all of these faults would be associated with a similar ~2 ka recurrence time to that of the PIF, then extensional slip on each fault could be sequentially triggered by the megathrust earthquakes that occur approximately every ~0.2 ka in this region. However, triggering would depend on the relationship between the normal fault geometry and the locus of the megathrust slip, as discussed below.

**Megathrust earthquakes and Pichilemu fault slip.** Megathrust earthquakes induce an instantaneous reversal in the polarity of the stress field in the upper plate. This change may induce the seismic rupture of crustal faults that are optimally oriented with respect to the new stress field e.g., refs. [60,61]. To explore possible rupture scenarios, we modelled the Coulomb failure stress (ΔCFS) induced by different slip distributions during megathrust earthquakes and considering the PIF as a receiver normal fault. The Maule earthquake involved slip within two subsegments at the northern and southern parts of the rupture[31], and induced positive ΔCFS values of 1.3–6 MPa in the PIF area (Fig. 6A). Model results based on geophysical and paleo-seismological observations along convergent margins suggest that the heterogeneous frictional properties of the subduction megathrust may result in temporally variable slip patterns e.g., refs. [62]. It is therefore possible that these two Maule subsegments may be associated with different slip behaviour and stress transfer to the PIF. For this reason, we generated synthetic megathrust-slip distributions at different locations along the Maule rupture zone (Fig. 6A–D and Fig. S14A–D). Slip along the southern Maule subsegment would result in a ΔCFS of −0.2 MPa at the PIF, inhibiting slip (Fig. 6D). Conversely, megathrust slip along the northern subsegment would result in a ΔCFS of 0.66 MPa in the

PIF area (Fig. 6B), promoting slip (Fig. 6B). We noticed that ΔCFS values on the PIF increase progressively as the locus of synthetic slip shifts northward, reaching a maximum of 0.23–0.34 MPa for friction coefficients of between 0.4 and 0.75 (Fig. S14A, B).

It has been suggested that the down-dip segmentation of megathrusts may control the magnitude and characteristics of subduction earthquakes and the activity of crustal faults[63,64]. We compared the ΔCFS induced by earthquakes at variable megathrust down-dip depths by simulating synthetic slip scenarios (Fig. S14C). We observed that when the locus of slip is in the deeper part of the megathrust (between 40 and 50 km depth), ΔCFS values in the PIF area are negative (−2 MPa), thus inhibiting slip (Fig. 6C). In contrast, when the locus of slip occurs at shallower depths or directly below the PIF (20–30 km depth), ΔCFS values on the PIF are above 0.2 MPa, and thus promote slip (Fig. S14D). However, megathrust slip at depths of less than ~15 km reduced the ΔCFS values below 0.2 MPa (Fig. S14D), indicating a ΔCFS that is still positive, but less favourable for triggering slip along PIF.

We explore the potential triggering of PIF during the interseismic phase using the degree of plate locking estimated from GPS measurements obtained over the decade preceding the Maule earthquake[31]. The interseismic phase induced negative ΔCFS values at the PIF (−1.44 MPa, Fig. 6E) inhibiting normal fault slip. This is consistent with the lack of extensional crustal earthquakes in the area during the 25 years prior to the Maule earthquake[34], suggesting that normal faulting along the PIF is unlikely during the interseismic phase. Farías et al.[4] suggested that Mesozoic N-S-striking structures, such as the Vichuquén Fault (Fig. 1A), are favourably oriented for reverse reactivation during interseismic contraction and could contribute to the long-term build-up of topography. However, these N-S-striking faults are not included in the Chilean Database of Active Faults[58] and we found no field evidence supporting their potential Quaternary activity. Further studies are clearly needed to assess the influence of interseismic contraction in the reactivation of upper-plate faults.

The Coulomb stress models show that different slip distributions during megathrust earthquakes can either promote or inhibit slip along the PIF. These results may explain why the

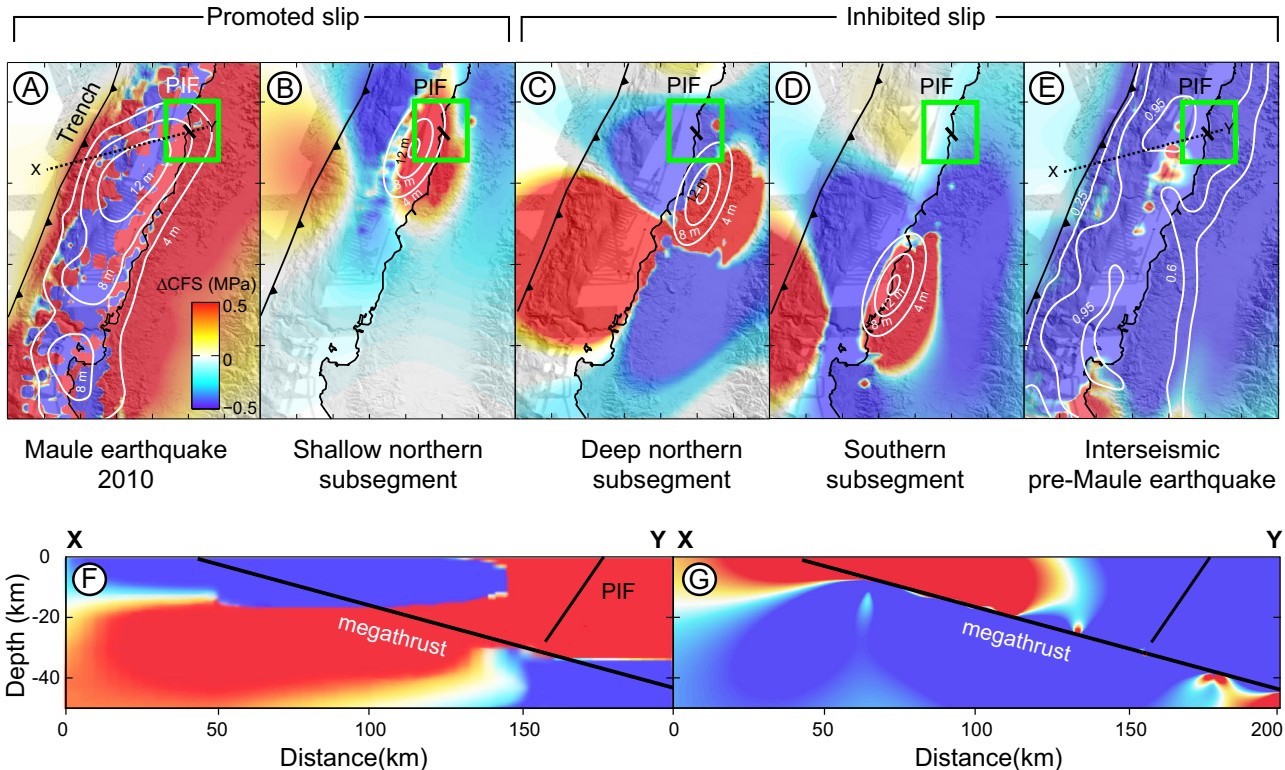

**Fig. 6 Coulomb stress changes and PIF slip.** Mean ΔCFS maps calculated for a depth range between 5 and 25 km using the fault parameters of the PIF and a friction coefficient of 0.4. **A** Mean ΔCFS map derived from the coseismic slip of the Maule earthquake[31] on the PIF. **B**–**D** Mean ΔCFS produced by synthetic slip scenarios and the same fault parameters as used in **A**. **E** Mean ΔCFS calculated using the degree of interseismic plate locking estimated from GPS measurements obtained over the decade preceding the Maule earthquake[31]. Note that positive ΔCFS that promote normal slip on the PIF occur only in scenarios depicted in **A**, **B**. **F** and **G** Profiles across the PIF showing ΔCFS during the 2010 Maule earthquake and during the pre-Maule interseismic period. White contours in **A**–**D** denote coseismic slip magnitude and the degree of plate locking in **E**.

estimated 2.1 ka recurrence time for large PIF earthquakes exceeds the recurrence time of megathrust earthquakes in this region. We propose two possible scenarios: First, if the Maule segment was characterised by ruptures with different lengths, magnitudes, and slip distributions, then the PIF could have been triggered by only one in 10–12 megathrust earthquakes, and only when megathrust slip involved the northern Maule sub-segment at shallow depths (20–30 km depth). Second, Mw > 8 historical megathrust earthquakes such as the 1835 and 1751 events suggest that the Maule segment was characterised by similar earthquake ruptures[65]. Previous studies of crustal faults in northern Chile have proposed that build-up of tensional stresses after successive seismic cycles may be counterbalanced by negative ΔCFS during the interseismic phase[61], extending the recurrence time of crustal faults and providing an explanation for the long recurrence times that we estimated for the PIF earthquakes.

**Origin of the PIF and implications for blind faults.** Blind faults usually occur in sedimentary basins when fault-tip propagation fails to reach the surface across a thick sedimentary cover[66]. This is, however, not the case for the PIF, which mostly affects crystalline basement rocks. Blind structures can also be controlled by mechanical and rheological heterogeneities in the upper crust[66]. The PIF host rocks are part of a Palaeozoic accretionary wedge characterised by pervasively deformed high-pressure and high-temperature metasedimentary and metavolcanics rocks (Fig. S1A) that locally form melanges[27–29,38]. These rocks have been over-printed by brittle deformation during Cenozoic exhumation, resulting in a rheologically heterogeneous and fragmented upper

crust[28]. We propose that this rheological heterogeneity may have preconditioned blind faulting by favouring strain diffusion across a broad zone of deformation within the uppermost crustal levels, thus preventing localised surface faulting. This assessment is further supported by the distribution of aftershock seismicity following the 2010 earthquakes across a ~10 km-wide zone (Figs. 1C and S7), and by low P-wave velocity inferred from seismic tomography, which defines an area characterised by intense fracturing[46]. We suggest that the lineaments observed in the LiDAR topography (Fig. 2) represent secondary fault-tip bending structures inherited from old crustal fabrics that might control their oblique orientation with respect to the PIF, but that are now either inactive or characterised by very low displacement rates.

The possibility that heterogeneities in upper-crustal rheology either favour or hinder surface-breaching faults is further highlighted by the characteristics of the El Yolki Fault, located 125 km farther south, which shares similar kinematics and orientation with the PIF[59]. In contrast to the PIF, the El Yolki Fault breaches the surface and is surrounded by metamorphic basement rocks consisting exclusively of moderately deformed metapelites displaying subhorizontal foliation[28]. This more homogeneous rheological character may therefore have promoted localized strain along the fault zone and towards the surface.

Unlike the megathrust, crustal-scale faults in the upper plate of subduction zones can remain inactive for long periods and in most cases not detectable by geodetic or seismic monitoring[67]. Furthermore, as some of these faults may be blind and lack clear geomorphic expression (such as the PIF), their unambiguous identification by field observations or remote-sensing methods is

challenging. Many of these cryptic crustal faults at subduction margins were only recognised following their seismic reactivation, which raises the question of how many other active faults with the potential to produce large earthquakes may exist in these environments. For example, a statistical analysis of crustal earthquakes in New Zealand has shown that more than half of the historical large earthquakes ruptured along previously unidentified faults[1]. Further cryptic and partly hidden active faults have also been documented at the Cascadia[68,69] margin as well as in the Kanto region of Japan[70], and similar conditions are likely to exist along other subduction zones. We conclude that regional-scale acquisition of LiDAR topography, combined with the quantitative analysis of off-fault and on-fault geomorphic markers and geodetic data is a promising way forward that will ultimately improve our knowledge of the location, deformation mechanisms, and recurrence time of hitherto unidentified seismogenic faults at convergent margins. The approach proposed in this study to detect blind faults and recognise their potential as cryptic seismogenic sources will contribute to the development of more accurate active-fault maps and hazard assessments, with far-reaching implications for seismic risk management in coastal areas.

## Methods

**Estimating coseismic slip during the PIF earthquakes.** We processed two Envisat® radar scenes acquired 2 days before and 7 days after the March 11 2010 earthquake doublet, obtaining line-of-sight (LOS) displacements (Fig. S8A–F, see details on the radar images and interferogram in Table S2). Because the interferometric coherence is relatively low (Fig. S8B), we applied filtering algorithms based on a moving window using an adaptive range filter[71] to facilitate the unwrapping (Fig. S8C). The interferograms were processed using Roi-Pac software[72] and unwrapped using the branch-cut algorithms. To corroborate our results, we compared the InSAR LOS displacement with those estimated using the permanent GPS station PICH projected to the LOS vector (location in Fig. 3D). To estimate coseismic slip along the PIF, we searched for the parameters that best reproduced the distribution of LOS displacements using forward elastic dislocation modelling[73]. The elastic dislocation models were programmed in Matlab® using the function okada85©[74]. We inferred fault geometries from alignments in the cluster of crustal aftershocks associated with the PIF earthquakes (55° dip, N38°W strike and 26 km length for F1 and 72° dip, N16°W strike and 16 km length for F2; Fig. S7). The strike inferred from aftershocks is similar to those from focal mechanisms[34] and interpretations based on aftershock distributions[34,46,75]. Aftershocks extended continuously down-dip from between ~2 and ~6 km depth to the megathrust at ~26 km (Fig. S7); however, the up-dip limit is rather diffuse. We used forward modelling to estimate the slip and up-dip depth of each fault for defined ranges and constant increments (Table S1). In addition, we included a hypothetical up-dip scenario of surface-breaching faults (Fig. S13). We set the range of slip values from 0.6 to 1.5 m for F1 and 0 to 0.65 m for F2, respectively, based on previous fault-slip inversions[34]. Using these ranges, we generated 85,918 elastic models (Table S1 and Fig. S9A) and searched for the parameters that minimised the NRMSE.

The root mean squared error (RMSE) is defined as the difference between observation (yi) and model (y), with n being the number of observations (Eq. 1).

$$\text{RMSE} = \sqrt{\frac{\sum_{i=1}^{n}(\text{yi} - y)^2}{n}} \qquad (1)$$

The NRMSE facilitates comparing models with different scales by normalising the RMSE (Eq. 2), where y max-y min is the range of observations. The uncertainties in model results were estimated using the lower 5% tail of the NRMSE distribution (Fig. S10). We estimated slip values using this tail distribution and defined the model uncertainty as the interval between the 5 and 95% percentiles (Fig. S10).

$$\text{NRMSE} = \frac{\text{RMSE}}{(\text{ymax} - \text{ymin})} \qquad (2)$$

To evaluate the consistency of our coseismic slip estimates we compared results from our forward model with those of an inverse model for a heterogeneous slip distribution. The inverse model was based on the automated fault model discretization method of Barnhart and Lohman[76], which can resolve scattered surface displacement observations by varying the model resolution with depth. We resampled the LOS displacements based on a Delaunay triangulation algorithm allowing to generate a dataset tractable for fault-slip inversions[76] (Fig. S11A–C). We inverted the resampled interferogram using the geometry of fault F1, a rake of −90, a fault length of 57 km to avoid boundary artefacts, and a depth of 26 km to the intersection with the megathrust. The fault discretization method is based on

an iterative approach that resizes the triangular fault-slip patches according to the model resolution. This generates smaller patches near the surface and larger patches at depth and offshore. The resulting model geometry of F1 comprised 678 patches (Fig. S11D). The green function matrix calculated for the patches may lead to unstable solutions that can be resolved using regularisation. The regularisation parameter (lambda) is dependent of the data noise and the Laplacian smoothing. The most suitable regularisation represents the correct balance between smoothing and data noise to fit the underlying noise-free signal; while high or low regularisation values result in increasingly smooth or increasingly complex models that do not represent a good fit to the noise-free signal. We selected the best fit slip distribution using the L-curve method[77] that compares graphically the squared root of model and data misfits, and the optimum lambda is then determined at the point of maximum curvature (Fig. S11E). We additionally checked the jRi value as a function of the lambda and confirmed that the optimum lambda was selected in the jRi-low range[76] (Fig. S11F). The jRi value is a metric of the quality of the inversion and is related to the input data noise, the fault regularisation and fault parameters, therefore, a lower value of jRi represents a more suitable regularisation value and smoothing[76]. We finally analysed the distributions of slip and model misfits statistically by re-discretizing the triangular meshes into a grid of equal area cells. The contribution of each triangle on each cell was calculated using a triangle area weighted average, this procedure allowed reducing the spatial bias produced by triangles of different sizes.

**Analysis of on-fault geomorphic features.** We performed a detailed geomorphic and morphometric analysis of off-fault and on-fault geomorphic features in the PIF area. On-fault features were identified by analysing topographic and fluvial metrics using the Topotoolbox-2 software[35]. We created red-relief maps (RRMs)[78] that combine the terrain openness with surface slope. RRMs are useful for identifying lineaments and fault scarps together with changes in the fluvial network, because they lack the potential bias of light source direction common in shaded-relief maps (Fig. S1B). We also generated local relief maps (Fig. S2C) using a 500 m roving window, and slope maps (Fig. S2D) calculated using the 8-connected neighbourhood gradient algorithm of Topotoolbox®[35].

Drainage and catchment morphology have the potential to be used as tectonic markers in the quantification of regional strain, uplift, and tilting. Ideally, catchments should be symmetric about the main trunk stream if they have incised a horizontal surface of uniform lithology under homogeneous climatic conditions[79]. To estimate the catchment asymmetry, we extracted the main catchments, the drainage networks and the main trunk streams in the Pichilemu area using the flow-routing and accumulation algorithms of Topotoolbox®[35]. We subsequently estimated the catchment centreline by creating a distance buffer from the border of each catchment polygon. Then the distance matrix was skeletonised using plain curvature to obtain the centreline and was compared with the main trunk stream to estimate the catchment symmetry factor (Fig. S2B). The symmetry factor Ts[37] (Eq. 3) is used to highlight changes in catchment asymmetry and areas of lateral tilting and is defined as:

$$\text{Ts} = \text{Da/Dd} \qquad (3)$$

where Da represents the distance from the catchment centreline to the main catchment trunk and Dd is the distance from the catchment centreline to the catchment boundary. The Ts values range between 0 and 1, where 0 represents a perfectly symmetric catchment and 1 represents a tilted catchment. Da orientations are presented as vectors that are colour-coded by Ts and incorporated into rose diagrams to interpret the dominant tilting direction (Fig. S2B).

Knickpoints were identified using the knickpointfinder algorithm in Topotoolbox[80], which iteratively maps the location of knickpoints by fitting a concave upward profile to the river-profile elevations, with the condition that the fitted curve must run below the profile elevations. The maximum vertical distance between the observed and modelled profile is defined by the tolerance parameter. We used a tolerance of 18.5 m, based on the maximum difference between the upstream maxima and downstream minima of the main trunk river profile[80].

The steepness index (Ksn) can be useful for obtaining information on tectonic and/or climatic perturbations in a fluvial network. Ksn values allow the quantification of deviations from the steady-state concavity of a river profile and the detection of these deviations along the stream network. The estimation of Ksn values is based on the power law (Eq. 4) of detachment-limited incision into bedrock[81].

$$\frac{dz}{dt} = U - kA^m \left(\frac{dz}{dx}\right)^n \qquad (4)$$

Where U is the uplift rate, A is the upslope area, dz/dx is the channel slope and m, k and n are constants. In steady-state conditions dz/dt = 0; hence, we can rearrange the equation as [Eq. 5]:

$$\frac{dz}{dx} = \left(\frac{U}{k}\right)^{\frac{1}{n}} A^{\frac{-m}{n}} \qquad (5)$$

were $(U/k)^{1/n}$ represents the channel steepness and m/n is the channel concavity

($\theta$); thus, the equation can be written as [Eq. 6]:

$$K_{sn} = \frac{S}{A^{-\theta}} \qquad (6)$$

In contrast to the $K_{sn}$, chi-plots are based on the horizontal transform of the upslope area to linearise the concave upward profile for a well-chosen reference concavity. This spatial transform makes chi-plots useful for identifying transient erosional signals with a common origin propagating upstream along the drainage network, such as tectonically generated knickpoints or changes in base level. Furthermore, the chi dimension allows comparing these signals between different catchments, irrespective of their size or shape.

Estimating chi-values requires rearranging Eq. [4] to convert dx to a distance measured from the catchment outlet and assuming that $U$ and k are spatially uniform (Eq. 7).

$$\int \frac{dz}{dx} dx = z(x0) + \left(\frac{U}{kA0^m}\right)^{\frac{1}{n}} \int \frac{A0}{A(x)^{\frac{m}{n}}} dx \qquad (7)$$

The integral of $dz/dx$ is calculated using a reference area (A0), where x0 is the catchment outlet. The chi-plot values ($\chi$) are estimated based on the right hand-integral (Eq. 8).

$$\chi = \int \left(\frac{A0}{A(x)}\right)^{\frac{m}{n}} dx \qquad (8)$$

To calculate the $K_{sn}$ and chi-values in the catchments of the Pichilemu area we used Topotoolbox®[35], which includes all the above equations. We used a reference concavity of 0.45.

**Analysis of off-fault geomorphic features.** To analyse off-fault geomorphic features we studied marine terraces using two methods, depending on the type of terraces and their origin, with the aim to identify and map marine shoreline angles. The shoreline-angle is a geomorphic marker located at the intersection between the paleo-platform and paleo-cliff that represent the maximum reach of the sea level during a highstand period that can be used to estimate vertical deformation and coastal uplift rates[40].

We analysed wave-built marine terraces following the principles outlined in Jara-Muñoz and Melnick[43] taking into account the morphology of the bedrock unconformity, the number of sedimentary cycles within the wave-built terrace, and the thickness of the sequence (Fig. S4). We mapped the surface morphology of the wave-built terraces using swath profiles to detect any breaks in slope. We also measured the depth to the crystalline bedrock in incised valleys and generated an isopach map of sedimentary sequence thickness (Fig. S4A, B). This allowed us to differentiate sedimentary-sequences and to improve the estimation of shoreline-angle elevations.

We studied the surface morphology of marine terraces using LiDAR topography and swath profiles in order to measure the locations and elevations of shoreline angles (see Supplementary Data 1). To estimate uplift rates ($u$), we correlated terrace levels with sea-level highstands using the IRSL ages and a composite sea-level curve for the southern hemisphere spanning the last 700 ka[44,45] (Eq. 9, Fig. S6).

$$u = \frac{(E - e)}{T} \qquad (9)$$

where $E$ is the elevation of shoreline angles, $e$ is the elevation of the corresponding highstand and $T$ the age of the terrace level. Uplift rate errors $Se(u)^2$ were estimated following Gallen et al.[82] as [Eq. 10]:

$$Se(u)^2 = u^2 \left(\left(\frac{\sigma^2 H}{H^2}\right) + \left(\frac{\sigma^2 T}{T^2}\right)\right) \qquad (10)$$

where $\sigma H$ is the error in relative sea-level, defined as [Eq. 11]:

$$\sigma H = \sqrt{\sigma E^2 + \sigma e^2} \qquad (11)$$

where $\sigma T$ is the age uncertainty in the sea-level curve (7 ka), $\sigma E$ is the error of the shoreline-angle assessments and $\sigma e$ is 12 m uncertainty of the highstand elevation based on Rohling et al.[45].

**Post-IR IRSL dating.** Four sediment samples were analysed by thermoluminescence dating were analysed at the University of Cologne using the post-IR IRSL signal of K-feldspar obtained from marine terrace sediments. The sedimentary units sampled were (1) deposited as shallow marine sediments, ideally in berm or swash-zone environments; (2) comprised of sandy sediments with medium grain sizes and more than 20% of feldspars and (3) from the base of the sequence, as close as possible to the bedrock wave-cut platform (Fig. S3). We analysed medium-sized K-Feldspar sand grains (100–250 μm) following the post-IR IRSL$_{290}$ SAR protocol[83]. The dated sediments were generally characterised by high feldspar signals with adequate reproducibility in dose-recovery tests performed after signal resetting in a solar simulator for 12 h (satisfactory ratios between measured and laboratory dose between 0.9 and 1.1). Burial doses were based on 5–19 aliquots of 8 mm diameter, using the central age model for calculation[84] (See Abanico-type plots in Fig. S5). In addition, we evaluated the completeness of signal resetting using a sample from a

modern beach berm (Sample SM mb, Table 1), which revealed an insignificant residual dose compared to the burial doses of the marine terrace samples (i.e., ~3.4 Gy). Radionuclide analysis (uranium, thorium and potassium) for dose-rate estimation was carried out using high-resolution gamma-spectrometry (see details in Table 1). We used a potassium content of 12.5 ± 0.5% to estimate internal dose rates, based on the method proposed by Huntley and Baril[85].

**Slip rate of the PIF from deformed marine terraces.** We estimated the PIF slip rate by searching for the best-fitting input parameters (i.e., the lowest NRMSE value) in a set of 116,964 model runs to find the minimum NRMSE value (Fig. S9B, C, Eq. 2). Uplift rates derived from shoreline-angle elevations of marine terraces were reproduced by forward elastic dislocation models by varying up-dip and down-dip slip depths, as well as the slip rate, of each fault (Table S1). The elastic model setups include the same geometry and down-dip depth used in the coseismic models. We performed preliminary modelling experiments using one and two faults, obtaining a better result using two faults (Fig. 4D). To perform the comparisons, we iterated the slip magnitude and up-dip depths of F1 and F2 using constant increments (Table S1). The best-fitting models are those with the lowest NRMSE value (Fig. S9B, C). To estimate the confidence interval of the best-fit models, we used the lower 5% tail of the NRMSE distribution (Fig. S10E, I). From these distributions, we defined the 5% and 95% percentiles as the confidence limits, equivalent to a 90% confidence interval, as the uncertainty of the best-fit model results (Fig. S10E–L).

Because marine terraces are exposed along the coast, they mostly reflect deformation along a 2D profile. To increase the area available for comparing dislocation-model results with observations we carried out a natural neighbour interpolation using a Delaunay triangulation of the scattered shoreline angles in 100 m bins (Fig. 4D). Because the spatial distribution of marine terraces is not suitable for estimating fault slip with a 3D inverse model, we used forward dislocation models in order to produce comparable results at both coseismic and long-term timescales.

**Modelling Coulomb stress failure.** We used the ΔCFS to evaluate the potential for a megathrust earthquake to induce slip along the PIF. We modelled the ΔCFS in the PIF area and directly on the PIF using the fault geometry inferred from aftershocks (Fig. S7) and the Coulomb 3.4 algorithms[86]. The model setup and rheological parameters were based on previous studies of the South American margin and in the Maule region e.g., refs. [4,34,61], we used a Poisson's ratio of 0.25 and Young modulus of 75 GPa both based on average values for upper-crustal materials[86]. The friction coefficient of crustal faults commonly varies over a wide range[61], and our analysis therefore considered a range of friction coefficients between 0.4 and 0.75.

Coulomb stresses imparted by slip on a source fault can either promote or inhibit slip along a receiver fault[60]. For this analysis we modelled the slip imparted during the 2010 Maule earthquake and synthetic slip scenarios in elliptical patches of 100 km length and 50 km width (Fig. S14A, C). We set the area and amount of slip of these synthetic slip patches based on the average slip distribution of the 2010 Maule earthquake. In a first set of modelling experiments, the locus of 28 synthetic slip patches was shifted along the Maule rupture zone between 38° and 35°S and within its up-dip and down-dip depth limits[31]. In a second set of experiments we shifted the locus of slip of 15 synthetic patches to different down-dip depths values of between 10 and 50 km, beyond the down-dip depth limit of aftershock seismicity[33]. We also modelled the potential effect of continental contraction induced by interseismic plate-locking on the PIF using a back-slip model based on the pre-2010 Maule earthquake locking distribution of Moreno et al.[31]. We considered a plate convergence rate of 66 mm/yr[26] and assumed complete elastic strain release during the 1835 earthquake, which was similar in rupture area and magnitude to the 2010 Maule earthquake[31].

## Data availability

The data needed to evaluate the conclusions in the paper are present in the paper and/or the Supplementary Information and Supplementary Data 1. Additional data related to this paper such as the LiDAR topography may be requested from the authors.

## Code availability

The Matlab® codes used for marine terrace mapping and dislocation modelling are part of TerraceM available at www.terracem.com.

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

## Acknowledgements

This study was supported by the Millennium Scientific Initiative (ICM) of the Chilean government through grant C160025 "Millennium Nucleus CYCLO The Seismic Cycle Along Subduction Zones", Chilean National Fund for Development of Science and Technology (FONDECYT) grants 1181479, 11180509 and 1190258, the ANID PIA Anillo ACT192169, and the MARISCOS (MAule eaRthquake: Integration of Seismic Cycle Observations and Structural investigations) project financed by the German Science Foundation (DFG), grant STR373/30-1. J.J -M. was supported by the DFG "LIFE" project, grant JA2860/1. M.S. was supported by the DFG "ARTE" project, grant STR373/41-1. LiDAR data was provided by Forestal Arauco to the CYCLO project. We would like to thank Kenneth Fisk and Christian Hillemann for help in the field and also Richard Arthur, Alejandro Araneda, Sonia Muñoz Belmar, Andres Tassara and Javier Morin for support before and during the field campaign.

## Author contributions

J.J. -M., D.M., and M.S. designed the study, participated in the fieldwork, analysed the field observations and models. S.L. carried out the inverse modelling, J.C. -A. carried out the Coulomb modelling, A.S. processed and analysed the radar interferometry, D.B. analysed the IRSL samples. J.J.-M. and D.M. wrote the paper. All authors commented on the paper.

## Funding

## Competing interests

The authors declare no competing interests.
