## [Peer Review File · Nature Communications]

The cryptic seismic potential of the Pichilemu blind fault in Chile revealed by off-fault geomorphologyReviewers' Comments:

Reviewer #2:

Remarks to the Author:

The authors present an important study analyzing the earthquake rupture patterns of blind faults in subduction zones with reference to the Pichilemu fault in Chile which is an upper-plate fault that was activated shortly after the 2010 M8.8 Maule earthquake.

The authors combine observations of deformation over several timescales to (1) constrain the slip rate of the Pichilemu fault, (2) show that the fault is blind which has implications for its hazard, and (3) determine what behavior along the subduction zone fault triggers upper plate deformation.

Overall, I think this manuscript presents an interesting case of combining datasets to access the seismic behavior of cryptic fault and develop testing hypothesis (with enough time) of the upper plate and subduction zone fault behavior. It is well-written. Of the datasets that the authors use, my strongest expertise is in earthquake source inversions so many of my comments are directed to towards that part of the text. My largest concern relates to several technical aspects of coseismic slip model. Addressing these comments will give much more credibility to the paper and ultimately increase its impact. I recommend publication after the following are addressed.

Coseismic uplift: Overall, more detail is needed in the main manuscript for the reader to access the slip inversion.

Comments on the main text:

Clarify that you are doing a set of forward models. Are you inferring one magnitude of slip per fault?

Clarify the meaning of iterating up- and down-dip depth values. Do you move the whole fault up and down? What variations of slip and depth do you explore?

What do you mean that your results are indistinguishable from a least squares inverse model? I am not sure that this sentence is needed. It also sounds strange as individual forward and inverse models are different and distinguishable, even without valuing one type over the other.

For the long-term slip rate, what is the bound on the fault being blind? You include the depth of the top of the fault. What is the uncertainty in this parameter? Given that there is some loss of coherence and the model inversion does perfectly fit the data near the fault, it is important to put a bound on this parameter. In the supporting material, you could also show a forward model with a fault that rupture the surface. Assuming the fit to the data is bad, this would convince the reader that the fault must be blind.

Figure 4: It would be helpful to show the original interferogram. My guess is that there is some loss of coherence due to the high topographic relief and vegetation. This will allow readers to better access the quality of the fault model.

Line 96: Describe over what spatial scale and any other details relevant for the smoothing of the shoreline angles.

Line 351: indicate specifically that you made an interferogram.

Line 365: In English, the "." Should probably be a ",".

Line 377: This paragraph is confusing. In the rest of the text, it sounds like you are solving a forward problem, but here you indicate that you are also inverting for the slip distribution. Why do you do both? Much more detail is needed to describe the inverse problem.

Supporting Material:

Figure S7: This figure does not convince me that your slip model is able to reproduce the deformation signal. It is important to show that LOS displacements (Figure S7C) on the same color scale as the forward model (4a). The residual is S7D look big and spatial coherent and I do appreciate that the color scale is different than S7C. This is worrying to me. Showing the forward model with the same color scale and LOS displacements would help me better compare. Also show the residual including the sign, not just the absolute differences.

S7A: no units. S7C: Label Unwrapped. Caption is wrong- only goes to (C).

S9: In the fence diagrams, what was the spacing that you used to test parameters? It is hard to show how the 3D space was tested. Another or an additional way is to show with residual vs. parameter value for each of the tested parameters. Sometimes, this can be clearer as it shows the convergence of each parameter.

With the exception of the 2010 displacements, the landscape was relatively unchanged post Maule. I.e., the evidence discussed here was there if we knew how to look for it. With the insight from this study, can these methods be applied to other subduction zone environments to access the hazard from upper plate faults? What lessons does this study provide? This is somewhat addressed in the last paragraph but a little more discussion would be interesting.

Line-by-line comments:

Line 30: While not well-mapped, portions of the Ridgecrest rupture had been previously mapped.

Line 45: What do you mean that the state of stress changes in space? Are these small changes that do not involve a change in sign of the stress?

Line 72: Is subducting

Figure 3: Define MIS in the caption like you have define LOS.

4.2 section title; Make active voice: Megathrust earthquakes can trigger the Pichilemu Fault

269: These are large ranges of depths in explore. Can depth of the the Maule earthquake be used to limit the bounds of megathrust earthquakes?

Reviewer #3:

Remarks to the Author:

Please see the attached document for detailed comments on the manuscript.

Kind regards,
Dr Laura Gregory

Review of 'The cryptic seismic potential of blind faults revealed by off-fault geomorphology, Pichilemu, Chile'

This manuscript presents results from the Pichilenu Fault (PIF) – an upper crustal structure that hosted two large normal faulting earthquakes 11 days after the M_w 8.8 Maule earthquake on the Chilean subduction megathrust (in 2010). Normal faults on the upper plate have an (obviously) lower average slip rate than the megathrust, and host earthquakes as a result of changes in stress in the upper plate that occur over the megathrust earthquake cycle. There is a very interesting interplay between the megathrust and upper plate faults, which has been recognised at subduction zones around the world, and this sequence deserves to be well-studied as it provides information on how these upper-plate faults behave.

The authors investigate the longer-term geomorphic signal of the PIF using geomorphic analyses and luminescence dating, and they then compare the long-term pattern of uplift to co-seismic displacement, finally calculating the coulomb failure stress *change* that results from slip on the megathrust. They apply a wide range of techniques to study the fault that crosses timescales. I think this is a helpful approach, but it also makes it quite challenging to treat every aspect of this broad study appropriately.

I recommend fairly moderate revisions to both improve the clarity of the manuscript, and to better constrain the interpretation of their results. I think that some changes may need to be done in how they analysed and present the geodetic/CFS data – and this may require some more time and investigation. My main concerns are with clarifying different aspects of the manuscript and in how the data are interpreted in light of the whole earthquake cycle on the megathrust; both of these concerns are detailed below. First – the interpretation and data analyses is the most major change that may need to be done.

Interpretation

The authors model InSAR data covering the earthquakes, but what is different here from previous work (Ryder et al., 2012) who have a more rigorous treatment of the geodetic data? Their result is more complex and better defined than the slip distribution in Figure S11. I wonder if the slip distribution in Ryder et al., 2012, would be more appropriate to use for a detailed CFS change calculation?

There is a lack of detail on the CFS calculations in the manuscript – for example there is not a figure showing the pattern of CFS change as a result of the different rupture scenarios, and it's not clear if they have actually modelled the CFS change as a result of the Maule earthquake, or of a 'similar' event. The discussion is strange too – it is obvious the CFS change will be greater due to rupture of the subduction zone that is closer to the PIF. Is there any further insight from the CFS pattern? How does the stress change vary over the megathrust earthquake cycle – e.g. is the transition from compression to extension instantaneous in the upper plate or does it occur over some time?

The authors use of 'stress threshold' needs more justification. If earthquakes occurred at a simple stress threshold, we would be able to forecast them much better. As far as I am aware, this value is unknown and varies depending on the properties of the fault and the history of stress accumulation and release. I think it would be more interesting to discuss the pattern of stress changes on the PIF fault in 3D, but there aren't any figures with the CFS plotted in 2D or 3D.

The authors discussion of stress and strain in line 288-onwards is not clear. What is meant by a minimum amount of elastic strain in line 292? Strain can be extensional or compressional. Also CFS is

usually discussed as a stress change, not just the stress value – we can never know the absolute value because we do not know the history of stress accumulation. Negative CFS just means stress in the opposite sense of what the authors have modelled for the fault – that ‘negative’ stress still counts in the total stress budget of the fault. Line 300-304 sort of explains this better, but this paragraph needs clarification.

Overall, the CFS modelling seems very basic compared to other studies, and it would be better to properly model the CFS change due to the interseismic vs. coseismic (megathrust) earthquake cycle, and show the spatial changes in a map or projected onto the PIF.

Clarification

I’m not sure if it will fit with the journal guidelines, but it would be helpful to have a sentence in the introduction, e.g. in the final paragraph, describing what you have done (DEM analyses, Quaternary dating, and InSAR). The reader would be informed on what kind of analyses to expect in the results section.

Take care with the use of ‘slowly-slipping’ (e.g. line 51). Slow slip is a very specific type of fault behaviour (stable sliding), whereas what you mean here is low slip rate faults, or even low average slip rate faults. These faults slip very rapidly during an earthquake, then do nothing, but some faults slip slowly for a period of time.

It is tricky using the aftershock clouds to infer the extent of the faults. It is possible the top part of the faults did not slip in this event, but have previously slipped and the scarp has been eroded away in the high mountainous environment. There are several examples of faults that partially rupture in one event, and then following the event, another rupture occurs on an un-ruptured patch above, below, or next to the patch that ruptured in the first event (e.g. Elliott et al., 2011, GRL, <https://doi.org/10.1029/2011GL046897>).

Care needs to be taken around the word ‘triggered’ throughout the manuscript – this word can imply instantaneous triggering, which the PIF earthquakes were not, because they occurred days after the mega-thrust event. This is true of other normal EQ’s over thrust faults – they often occur during the months and years after the megathrust event. This implies that the mega-thrust makes it more likely for the PIF to rupture, but does not instantaneously ‘trigger’ it. The authors sometimes treat this properly in the discussion that follows this sentence (e.g. an earthquake does not always occur on PIF, and it has a longer recurrence interval), but it should be explicit from the beginning that what is observed here is not instantaneous triggering.

Related to this point, it may be important include a discussion that faults like PIF may not always fail in M 6.6 – 7.0 earthquakes. There could be smaller magnitude earthquakes on overlying normal faults that occur with greater frequency. Yes, the PIF events were large, but there must be other faults or other scenarios where the events may not be so large, and the magnitude/recurrence interval on longer timescales cannot be inferred by the terrace uplift rates/long term slip rates.

Also, it makes sense to me that section 4.2 should come before 4.1 – e.g. discuss the process first, and then discuss it in light of your results. The the authors could more carefully interpret their results in light of the coulomb stress changes. I do not insist on this change but I just suggest the authors consider it.

The word ‘reactivation’ is not very appropriate here – the PIF is clearly an active fault (e.g. not an ancient structure that gets ‘reactivated’ in a different tectonic phase). ‘Reactivate’ suggests that it was not active before 2010 – but the authors suggest that the long-term activity on the PIF causes

the uplift and change in morphology of terraces. It's more straightforward to use something like (L 278) '... slip distributions along the megathrust may promote earthquakes on the PIF' or just something more specific to the behaviour observed.

Minor Line comments

30: I would argue that the Kaikoura earthquake occurred on known faults, just in a complicated way. Also, the Christchurch earthquake occurred in 2011, not 2016. Christchurch is related to Darfield, and could be considered two events that are part of a sequence.

44 and on: In this paragraph you use the word 'may' a lot – this takes away from the impact of your study. It's fine to use 'may' sometimes, but it's overused here.

45: clarify – by space do you mean the 'spatial distribution' of stress, e.g. stress concentrations?

94-96: The grammar isn't quite right here – I don't think you need the word 'neither'.

103-105: Figure 2A contains several different lines, but it is not clear exactly what the 'subparallel lineaments' are on the figure – please include the label for 'lineament' in the key for Figure 2A.

131-133: The authors state: 'Furthermore, the distance between knickpoints and lineaments is rather uniform (Fig. 2B), suggesting that knickpoints are not related to active fault scarps, and rather reflect base level changes associated with relative sea-level variations.'

However Figure 2C shows the distance between the knickpoints and sea level, not the distance to the lineaments. I would expect a uniform distance between the knickpoint and the lineament to indicate that the lineament is controlling the location of the knickpoint – because the lineament is not oriented in the same direction as the coastline/baselevel. Can the authors clarify this statement? From the map view, it does not look like the knickpoints are at an equal (uniform) distance from lineaments.

192-194: The authors state the faults 'offset the deformation pattern inferred from marine terraces' – is this the deformation pattern associated with the large-scale subduction zone deformation? It would be better to be specific here, and clarify how the uplift of terraces due to the PIF has been untangled from the deformation of the coast due to the megathrust.

194-197: Is there a figure that shows the two models?

219-221: These two observations are in agreement with one another – there is no need to use the word 'whereas'.

365: 68,921 models?

Figures:

Figure 1: What magnitude is the aftershock catalogue complete to? It would be helpful to have a sense of what magnitudes are plotted on this figure.

Figure 2: The label 'edge of seismicity' is a little confusing because it looks like it is pointing to the two dashed lines (but the seismicity occurs beyond these according to figure 1). This could be clarified as 'NW boundary of seismicity'. It's not clear what the heavy dashed black lines represent – these, along with the 'lineaments' should be included in the key because the various annotations on this figure are confusing.

Figure 4: I suggest using a different colour scales for (a) and (d). The colour scales are similar but different – in that the same colours are used but in a different order, which is strange, and they are showing two different processes (MIS stage vs LOS), and there is no need for them to be the same.

Figure 5: It's not clear what the purple boxes represent in this figure. It would be better to include a 2D or 3D plot of the CFS change.

Supplement

Figure S11 – there are some very strong residuals in the S11d – what is being missed in the modelling?

Figure S12 – which of these scenarios is closed to the Maule earthquake – and what is the resulting pattern of CFS change?

Reviewer #4:

Remarks to the Author:

This is a review of "The cryptic seismic potential of blind faults revealed by off-fault geomorphology, Pichilemu, Chile". This manuscript presents novel results on blind active faulting on the Pichilemu Fault in the upper plate of the subduction zone, central Chile. The study nicely integrates several different datasets involving field work, remote sensing analysis, and modeling. The study analyzes on- and off-fault geomorphology using morphotectonic and fluvial network analysis, uplifted marine terraces, and luminescence dating, determines the total slip and fault geometry using InSAR, seismicity, and modeling, and addresses reactivation potential of the fault during megathrust earthquakes using Coulomb failure stress modeling. The results indicate that the Pichilemu fault is a blind fault that has a slip rate of 0.42mm/yr and a recurrence interval an order of magnitude higher than the recurrence intervals of megathrust earthquakes on the subduction zone. The findings are broadly applicable and will be of interest to those studying seismic hazards and active faulting in subduction zones (as well as seismic hazards in general), in addition to those studying the tectonics of Chile or the Andes. The conclusions are generally supported by the observations and interpretations presented in the manuscript but do require some further clarification and comparison to previous similar studies.

As I am not an expert in the modeling aspects of this study, my review largely focuses on the tectonic geomorphology methods, results, and interpretations. Below I outline several major issues that should be addressed to help strengthen the message and clarify the methods and interpretations for readers. I also include a list of minor comments by line number, largely to clarify statements in the text.

Major issues include:

1. The interpretation of the lidar that demonstrates the "on-fault" tectonic geomorphology requires additional documentation or explanation to convincingly show or describe possible surface deformation. For example, for the lidar interpretation, there are several other linear features that extend farther than the mapped lineaments, but these are not mapped as lineaments. Aligned linear drainages extend farther northwest from one of the lineaments shown in Fig. S1B. The lineament shown in Fig. S1C is also not convincing, at least as shown. Also note that the white arrows depicting the ends of lineaments in Fig. 2E do not match the extents of the black lines in Fig. 2A and Fig. S1B. The red lines drawn in Fig. 2F do not match with the lineaments shown in Fig. 2A or 2E. It appears that the top 2 profiles (1 and 2) should align with the lineament shown in 3, with another lineament immediately to the NE. Finally, it is not clear what "distance to lineament" on the x-axis represents – perhaps label on Fig. 2E for clarity?
2. The relationship between the lineaments mapped at the surface, the zone of high seismicity depicted in Fig. 2, and F1 and F2 isn't clearly described. The lineaments are oblique to F1 and F2, and it is unclear how they represent "on-fault" tectonic geomorphology, especially if the fault zone is determined to be a distributed zone that is ~10 km wide. In addition, it is also unclear what "Footwall" and "Hanging wall" in Fig. 2A refer to – are these dashed lines the footwall and hanging wall only for F1? It appears from comparing Fig. 3 with Fig. 2 and Fig S1 that the Fault F1 lies between the dashed hanging wall and footwall lines, but then the zone of high seismicity appears to define F1 and F2 in Fig. 1C. Perhaps these relationships could be clarified in a figure or in the text?
3. The interpretation of the catchment and drainage metrics that demonstrate the "on-fault" tectonic geomorphology requires additional documentation or explanation to convincingly show or describe variable surface deformation. The catchment asymmetry and alignment of channels/elongated basins that are subparallel to the dashed lines in Fig. 2 is perhaps more convincing evidence of a zone of blind faulting being expressed at the landscape rather than the oblique lineaments, but these should be described in more detail. The catchment asymmetry would be more convincing if the numbers for the asymmetry factor were presented and discussed. In addition, the easternmost part of C2 appears to be located in the footwall, a part of the landscape presumably uplifting, yet there are no knickpoints or higher channel steepness in that region. How do the authors explain the lack of evidence of uplift in this part of the catchment, if the presence of knickpoints, increased relief, and higher channel steepness is used to explain uplift of C1? This discussion of the differences between the two catchments would be strengthened if more quantitative metrics (instead of qualitatively discussing

observations) were discussed to clearly show that one catchment may be uplifted. For example, what about drainage basin relief or relief ratio, drainage basin slope, or some other indices of tectonic activity? Are there differences between the two catchments using some of these metrics?

4. The bedrock lithology and any variations in precipitation over the catchments and their effect on the tectonic geomorphology is also unclear. Early in the manuscript, the bedrock is described as crystalline metamorphic bedrock and assumed to be homogeneous. Later in the manuscript (L308-323), the bedrock is described as more heterogeneous metasedimentary and metavolcanics units with an overprint of Cenozoic brittle faulting. If the latter description is more accurate, what effect does that have on the catchment and drainage metrics used to infer deformation? For example, are any of the knickpoints or zones of higher channel steepness associated with any of these subtle changes in the bedrock lithology? In Figure 2, there appears to be several prominent NNE-SSW ridges at the southern end of C2 (one of them corresponding to a knickpoint) and just west of C2, in addition to a NE-SW elongate drainage pattern on the SW side of C2. Is this controlled by the bedrock lithology or Cenozoic brittle faulting? Regarding the climate, is there any variation in the precipitation or vegetation over the catchments that may impact how the fluvial network responds?

5. There appear to be minor inconsistencies in the marine terrace mapping presented in Fig. 3a and the profiles shown in Fig. S4, namely, the mapping and extent of marine terrace MIS5c/e. In Fig. 3a, the relatively flat, uplifted area NE of F1 (between "Dune Fields" label and the white star) isn't mapped as a marine terrace, yet in the profiles in Fig. S4, this area is mapped as MIS5e. Could the authors clarify or revise for consistency?

6. There have been previous studies (e.g., Farias et al., 2011, Tectonics – cited in this manuscript but not discussed in detail) that discuss the geometry of the Pichilemu fault from seismicity and geomorphology and use Coulomb stress analysis to discuss triggering and activity of the fault. This manuscript would greatly benefit from discussing how their results are similar to or different from previous similar work on the geometry and activity of the Pichilemu fault to improve our understanding of the fault and its associated seismic hazard.

Minor Comments:

L104: These lineaments appear to be oriented obliquely to the zone of high seismicity, but the relationship between these lineaments, the zone of high seismicity, and F1 and F2 isn't clearly described. I see this is briefly addressed in L320-323, but this statement appears to be highly speculative as no detailed information on the bedrock lithology heterogeneity nor how or why these secondary fault tip bending structures are at such oblique angles to F1 and F2.

L117-L120: These statements would be clearer if they referenced specific features in C1 and C2 that demonstrated differences in the hanging wall and footwall. The higher relief and steeper channels in C1, interpreted to be in the uplifted footwall vs the lower relief and lower channel steepness indices in C2 in the hanging wall, for example. As mentioned above, this discussion would be significantly more convincing if quantitative metrics were presented and discussed.

L119: It is unclear how knickpoints were defined and picked, please add to this section or to the appropriate place in the methods section.

L123: in line 105, it states that there are NW-SE subparallel lineaments associated with slope breaks, but here in line 123, it says no slope breaks were observed. Please clarify.

L130: If the distance between the knickpoints and lineaments is uniform, why wouldn't that suggest a relationship to the active faulting?

L131: The relationship of the knickpoints and the distance to the lineament isn't clear if it's along-channel distance or measured in a straight line?

L311-313: In this paragraph, the description of the bedrock, previously characterized as homogeneous, appears to say that is the bedrock is heterogeneous, with different sections of metasedimentary and metavolcanics rocks that have an overprint of Cenozoic exhumation. Could the authors clarify the bedrock lithology? Along these lines, a geologic map in the supplementary material could be a useful addition.

L365: As I am unfamiliar with this modeling technique, how is it possible to generate a fraction of a model? Perhaps specify or clarify for readers less familiar with this technique?

L389: The only metrics presented in the manuscript include catchment asymmetry, channel

steepness, and chi values – if the authors used other metrics that did not show anything, perhaps that should be clarified here. The asymmetry factor also isn't presented – the observations of catchment asymmetry appears to be qualitative. If the asymmetry factor was calculated, that should be mentioned.

L398: Is there any difference in precipitation across the catchments, from orographic precipitation, for example? Or is the assumption of uniform climate (and vegetation) an accurate one?

Table 1. I suggest listing the total dose rate in Table 1, too.

We appreciate the effort and time that the reviewers have dedicated to our manuscript. We agree with most comments and suggestions, which you will find addressed in the responses and this thoroughly revised version of the manuscript.

REVIEWER COMMENTS

Reviewer #2 (Remarks to the Author):

Overall, I think this manuscript presents an interesting case of combining datasets to access the seismic behavior of cryptic fault and develop testing hypothesis (with enough time) of the upper plate and subduction zone fault behavior. It is well-written. Of the datasets that the authors use, my strongest expertise is in earthquake source inversions so many of my comments are directed towards that part of the text. My largest concern relates to several technical aspects of coseismic slip model. Addressing these comments will give much more credibility to the paper and ultimately increase its impact. I recommend publication after the following are addressed.

Coseismic uplift: Overall, more detail is needed in the main manuscript for the reader to access the slip inversion.

We decided to remove the section on slip inversion because it was complementary to forward modelling and was not directly used in the estimates of recurrence time. We believe the inverse model was redundant and added unnecessary complexity. Instead, we improved the description of the forward dislocation modelling by adding technical aspects of the coseismic slip model such as the ranges of parameters used for forward dislocation models and modelling experiments using blind and surface breaching faults.

Comments on the main text:

1) Clarify that you are doing a set of forward models. Are you inferring one magnitude of slip per fault?

We made a set of forward models. This clarification has been added to the first paragraph of section 3.3.

2) Clarify the meaning of iterating up- and down-dip depth values. Do you move the whole fault up and down? What variations of slip and depth do you explore?

We changed the fault width by looping the up-dip depth limit of slip, as explained in Section 3.3. In addition, we added details on the ranges of slip and up-dip depth limit in Table S1 and Section 5.1. We defined the down-dip depth range based on the aftershock seismicity (see details in the next answers).

3) What do you mean that your results are indistinguishable from a least squares inverse model? I am not sure that this sentence is needed. It also sounds strange as individual forward and inverse models are different and distinguishable, even without valuing one type over the other.

As explained in the beginning, we excluded the slip inversion from this study including only forward elastic dislocation models.

4) For the long-term slip rate, what is the bound on the fault being blind? You include the depth of the top of the fault. What is the uncertainty in this parameter? Given that there is some loss of coherence and the model inversion does perfectly fit the data near the fault, it is important to put a bound on this parameter.

We included details in the range of error of slip and up-dip for all faults in Section 3.3 and figure S10.

5) In the supporting material, you could also show a forward model with a fault that rupture the surface. Assuming the fit to the data is bad, this would convince the reader that the fault must be blind.

This is a very good suggestion, thank you! Following this comment, we included Figure S11 in the Supplementary Materials comparing our results with a surface breaching rupture.

6) Figure 4: It would be helpful to show the original interferogram. My guess is that there is some loss of coherence due to the high topographic relief and vegetation. This will allow readers to better access the quality of the fault model.

We included a new Figure S8A showing the original interferogram as well as a coherence map (S8B), and added more details on the steps of the InSAR processing, including information on the treatment of low-coherence areas.

7) Line 196: Describe over what spatial scale and any other details relevant for the smoothing of the shoreline angles.

The smoothing was done using natural neighbor interpolation using based on Delaunay triangulation over 100 m bin size. This mainly reduces the uncertainty associated with local effects such as bedrock heterogeneity and post-erosional deposition. We included more information on the interpolation methods in Section 5.3.4

8) Line 351: indicate specifically that you made an interferogram.

We modified this as suggested.

9) Line 365: In English, the “.” Should probably be a “,”.

We corrected this typo, using “,” as thousands separator here and along the text.

10) Line 377: This paragraph is confusing. In the rest of the text, it sounds like you are solving a forward problem, but here you indicate that you are also inverting for the slip distribution. Why do you do both? Much more detail is needed to describe the inverse problem.

We decided to remove the slip inversion, which had the aim to complement the forward modelling results. We believe that the inversion model added unnecessary complexity and basically the same results as the forward models. We used the spared space in describing more details of the forward model, such as range of parameters (Table S1) and forward model experiments (Fig. S11).

Supporting Material:

11) to show that LOS displacements (Figure S7C) on the same color scale as the forward model (4a). The residual is S7D look big and spatial coherent and I do appreciate that the color scale is different than S7C. This is worrying to me. Showing the forward model with the same color scale and LOS displacements would help me better compare. Also show the residual including the sign, not just the absolute differences. Figure S7: This figure does not convince me that your slip model is able to reproduce the deformation signal. It is important

We thank the reviewer for these suggestions. We made an effort to improve our model setup and managed to reduce the large residuals in the areas pointed out by reviewers #2 and #3. This was achieved by increasing the number of iterations and fixing the fault dip angle to 55°, based on a new relocated catalogue of aftershock seismicity that was published during the revision (Calle-Gardella et al. 2021). We excluded the iteration of the down-dip depth limit, as it can be directly estimated from the aftershock distribution, and is basically at the intersection with the megathrust. In addition, we increased the size of the modelling domain and thus area for comparison between LOS and model

displacements, and we also incorporated the secondary fault F2. Our results are now shown with the same color-scale and residuals include the sign.

Or results do not differ from the previous ones; the main difference is that now we include 10 cm of slip in fault F2, which agrees with the results of Ryder et al. (2012) obtained using an inverse model.

In the vicinity of the fault (<10 km), residuals are ca. 5 cm. We now point out that the residuals in the far-field that reach a maximum of 10 cm may be a result of afterslip and other post-seismic processes that followed the Maule earthquake, and were not accounted for in our modelling. Our relatively simple model reproduces the surface deformation with relatively low residuals and similar results to previous studies though using a more precise fault geometry.

The main conclusions of our study remain unaltered by the new modelling results.

12) S7A: no units.

We included the units in the revised Fig. S8.

13) S7C: Label Unwrapped.

This figure was modified and labels removed; we include “Unwrapped” in the caption of the revised Fig. S8

14) S7A: Caption is wrong- only goes to (C).

This caption was corrected.

15) S9: In the fence diagrams, what was the spacing that you used to test parameters? It is hard to show how the 3D space was tested. Another or an additional way is to show with residual vs. parameter value for each of the tested parameters. Sometimes, this can be clearer as it shows the convergence of each parameter.

We added the plots of all the parameters v/s NRMSE in Fig. S9. We included the fence diagrams as they allow a combined view of the parameters. For clarity we added detailed information on the parameters and their spacing intervals to the new Table S1.

16) With the exception of the 2010 displacements, the landscape was relatively unchanged post Maule. I.e., the evidence discussed here was there if we knew how to look for it. With the insight from this study, can these methods be applied to other subduction zone environments to assess the hazard from upper plate faults? What lessons does this study provide? This is somewhat addressed in the last paragraph but a little more discussion would be interesting.

We complemented the lines in the end paragraph including how combining different datasets and techniques will ultimately improve our knowledge of unidentified blind faults and the contribution to active faults maps and hazard assessments.

Line-by-line comments:

17) Line 30: While not well-mapped, portions of the Ridgecrest rupture had been previously mapped.

We replaced by “partly mapped faults” in this sentence.

18) Line 45: What do you mean that the state of stress changes in space? Are these small changes that do not involve a change in sign of the stress?

We rewrote this line changing space by “spatial distribution of stresses”.

19) Line 72: Is subducting

This typo was corrected.

20) Figure 3: Define MIS in the caption like you have define LOS.

MIS definition was added to the caption.

21) 4.2 section title; Make active voice: Megathrust earthquakes can trigger the Pichilemu Fault

We changed this title as: "4.2 Megathrust earthquakes and slip along the Pichilemu Fault"

22) 269: These are large ranges of depths in explore. Can depth of the Maule earthquake be used to limit the bounds of megathrust earthquakes?

We use the downdip depth of the Maule earthquake from the slip distribution of Moreno et al. (2012). We also added more information regarding the selection of depth ranges in the methods section of CFS. Part of this figure was changed and moved to Supplementary material following the suggestions of reviewer #3.

Reviewer #3 (Remarks to the Author):

I recommend fairly moderate revisions to both improve the clarity of the manuscript, and to better constrain the interpretation of their results. I think that some changes may need to be done in how they analysed and present the geodetic/CFS data – and this may require some more time and investigation. My main concerns are with clarifying different aspects of the manuscript and in how the data are interpreted in light of the whole earthquake cycle on the megathrust; both of these concerns are detailed below. First – the interpretation and data analyses is the most major change that may need to be done.

The authors model InSAR data covering the earthquakes, but what is different here from previous work (Ryder et al., 2012) who have a more rigorous treatment of the geodetic data? Their result is more complex and better defined than the slip distribution in Figure S11. I wonder if the slip distribution in Ryder et al., 2012, would be more appropriate to use for a detailed CFS change calculation?

In contrast to the data used by Ryder et al., our radar images cover a shorter time span before and after the PIF earthquakes (2 and 7 days before and after the PIF earthquakes, respectively, in contrast to 2 and 44 days before and after the earthquakes by Ryder et al. 2012). Therefore, the InSAR displacements probably include less effects from post-seismic deformation associated with the Maule earthquake.

Ryder et al. (2012) modelled fault slip distribution using an inverse model. Instead, we use forward elastic models to estimate both the long-term slip rate and the co-seismic to allow for a straightforward comparison. We adopted this approach as the 2D nature of the long-term uplift data precludes to use a 3D inverse model. However, our results for coseismic slip are analogue to those of Ryder et al. (2012).

To model the CFS change we use the slip distribution of the 2010 Maule earthquake of Moreno et al. (2011), which includes the most complete dataset (costal land-level changes, 169 GPS displacements, and InSAR). Instead, Ryder et al. (2012) only used InSAR data from Tong et al. (2010) to estimate slip during the Maule earthquake.

1) There is a lack of detail on the CFS calculations in the manuscript – for example there is not a figure showing the pattern of CFS change as a result of the different rupture scenarios, and it's not clear if they have actually modelled the CFS change as a result of the Maule earthquake, or of a 'similar' event.

We modified Section 4.2 including a new figure with the CFS patterns during the Maule earthquake, during the interseismic period before the Maule earthquake and three synthetic scenarios (Fig. 6). We moved the previous figure to the Supplementary materials.

2) The discussion is strange too – it is obvious the CFS change will be greater due to rupture of the subduction zone that is closer to the PIF. Is there any further insight from the CFS pattern? How does the stress change vary over the megathrust earthquake cycle – e.g. is the transition from compression to extension instantaneous in the upper plate or does it occur over some time?

The shift from contraction to compression is instantaneous as suggested by displacements from continuous GPS that recorded the earthquake (e.g., Moreno et al., 2012). We now include a discussion on CFS during the interseismic period and on the previous interpretation of Farias et al., 2011 regarding extension and compression during the different phases of the seismic cycle.

3) The authors use of 'stress threshold' needs more justification. If earthquakes occurred at a simple stress threshold, we would be able to forecast them much better. As far as I am aware, this value is

unknown and varies depending on the properties of the fault and the history of stress accumulation and release.

We rewrote section 4.2 removing the stress threshold and focusing instead on discussing the polarity of CFS.

4) I think it would be more interesting to discuss the pattern of stress changes on the PIF fault in 3D, but there aren't any figures with the CFS plotted in 2D or 3D.

We included in the new figure 6 2D maps of CFS.

5) The authors discussion of stress and strain in line 288-onwards is not clear. What is meant by a minimum amount of elastic strain in line 292?

We rewrote the Coulomb stress section (4.2) removing the threshold and focusing the discussion on the polarity of CFS changes.

6) Strain can be extensional or compressional. Also CFS is usually discussed as a stress change, not just the stress value – we can never know the absolute value because we do not know the history of stress accumulation. Negative CFS just means stress in the opposite sense of what the authors have modelled for the fault – that 'negative' stress still counts in the total stress budget of the fault. Line 300-304 sort of explains this better, but this paragraph needs clarification.

Indeed, the reviewer is right in and now we focused the discussion on the meaning of CFS polarity changes.

7) Overall, the CFS modelling seems very basic compared to other studies, and it would be better to properly model the CFS change due to the interseismic vs. coseismic (megathrust) earthquake cycle, and show the spatial changes in a map or projected onto the PIF.

We included a comparison of co-seismic (Maule earthquake) v/s interseismic CFS changes and map views of synthetic scenarios in figure 6 and Section 4.2.

8) I'm not sure if it will fit with the journal guidelines, but it would be helpful to have a sentence in the introduction, e.g. in the final paragraph, describing what you have done (DEM analyses, Quaternary dating, and InSAR). The reader would be informed on what kind of analyses to expect in the results section.

We agree and included a brief description of the methods used at the end of the introduction.

9) Take care with the use of 'slowly-slipping' (e.g. line 51). Slow slip is a very specific type of fault behaviour (stable sliding), whereas what you mean here is low slip rate faults, or even low average slip rate faults. These faults slip very rapidly during an earthquake, then do nothing, but some faults slip slowly for a period of time.

We changed "slow-slipping" to "low slip rate faults"

10) It is tricky using the aftershock clouds to infer the extent of the faults. It is possible the top part of the faults did not slip in this event, but have previously slipped and the scarp has been eroded away in the high mountainous environment. There are several examples of faults that partially rupture in one event, and then following the event, another rupture occurs on an un-ruptured patch above, below, or next to the patch that ruptured in the first event (e.g. Elliott et al., 2011, GRL, <https://doi.org/10.1029/2011GL046897>).

We agree with the reviewer. The up-dip terminations are diffuse instead the downdip terminations are clearer. We included this observation in the methods and section 3.3. Furthermore, we made new

forward models varying the up-dip termination and slip of both faults and fixing the downdip depth termination. The results remain mostly unchanged.

In addition, we indicate in Section 3.3 that the different up-dip terminations between the short-term and long-term forward models may be explained by partial rupture during the PIF earthquakes including the reference indicated.

11) Care needs to be taken around the word ‘triggered’ throughout the manuscript – this word can imply instantaneous triggering, which the PIF earthquakes were not, because they occurred days after the mega-thrust event. This is true of other normal EQ’s over thrust faults – they often occur during the months and years after the megathrust event. This implies that the mega-thrust makes it more likely for the PIF to rupture, but does not instantaneously ‘trigger’ it. The authors sometimes treat this properly in the discussion that follows this sentence (e.g. an earthquake does not always occur on PIF, and it has a longer recurrence interval), but it should be explicit from the beginning that what is observed here is not instantaneous triggering.

We explicitly indicate instantaneous and non-instantaneous slip triggering in the introduction to differentiate the activity of the Santa Maria and Pichilemu faults. We changed “trigger” in some parts of the discussion to avoid confusions.

12) Related to this point, it may be important include a discussion that faults like PIF may not always fail in M 6.6 – 7.0 earthquakes. There could be smaller magnitude earthquakes on overlying normal faults that occur with greater frequency. Yes, the PIF events were large, but there must be other faults or other scenarios where the events may not be so large, and the magnitude/recurrence interval on longer timescales cannot be inferred by the terrace uplift rates/long term slip rates.

From a regional survey of marine terraces along the Maule earthquake rupture zone, we found that several other active faults are associated with throw rates similar to the PIF and therefore likely also similar slip rates. Because most of these faults have normal kinematics, their slip is triggered by megathrust earthquakes, and of course may rupture during more frequent smaller earthquakes. We included these observations and hypotheses to the discussion in Section 4.1.

13) Also, it makes sense to me that section 4.2 should come before 4.1 – e.g. discuss the process first, and then discuss it in light of your results. The authors could more carefully interpret their results in light of the coulomb stress changes. I do not insist on this change but I just suggest the authors consider it.

We think the reader will appreciate a more intuitive way, going first to the results and then discussing the implications. However, we extended our interpretations of coulomb stress in the discussion.

14) The word ‘reactivation’ is not very appropriate here – the PIF is clearly an active fault (e.g. not an ancient structure that gets ‘reactivated’ in a different tectonic phase). ‘Reactivate’ suggests that it was not active before 2010 – but the authors suggest that the long-term activity on the PIF causes the uplift and change in morphology of terraces. It’s more straightforward to use something like (L 278) ‘... slip distributions along the megathrust may promote earthquakes on the PIF’ or just something more specific to the behaviour observed.

We replaced the word reactivation throughout the text.

Minor Line comments

15) 30: I would argue that the Kaikoura earthquake occurred on known faults, just in a complicated way.

Also, the Christchurch earthquake occurred in 2011, not 2016. Christchurch is related to Darfield, and could be considered two events that are part of a sequence.

Regarding the Kaikokura earthquake we now indicate specifically the Papatea fault, a previously unmapped structure that slipped during the Kaikokura earthquake (according to Hollingsworth et al. 2017). We corrected the years and grouped together Christchurch and Darfield as an earthquake sequence that ruptured previously unrecognized faults (according to Beavan et al., 2011).

16) 44 and on: In this paragraph you use the word ‘may’ a lot – this takes away from the impact of your study. It’s fine to use ‘may’ sometimes, but it’s overused here.

We modified this paragraph as suggested.

17) 45: clarify – by space do you mean the ‘spatial distribution’ of stress, e.g. stress concentrations?

We changed “spatial”, by “spatial distribution of stresses”.

18) 94-96: The grammar isn’t quite right here – I don’t think you need the word ‘neither’.

“neither” was removed.

19) 103-105: Figure 2A contains several different lines, but it is not clear exactly what the ‘subparallel lineaments’ are on the figure – please include the label for ‘lineament’ in the key for Figure 2A.

we added the lineaments to the key.

20) 131-133: The authors state: ‘Furthermore, the distance between knickpoints and lineaments is rather uniform (Fig. 2B), suggesting that knickpoints are not related to active fault scarps, and rather reflect base level changes associated with relative sea-level variations.’

However Figure 2C shows the distance between the knickpoints and sea level, not the distance to the lineaments. I would expect a uniform distance between the knickpoint and the lineament to indicate that the lineament is controlling the location of the knickpoint – because the lineament is not oriented in the same direction as the coastline/baselevel. Can the authors clarify this statement? From the map view, it does not look like the knickpoints are at an equal (uniform) distance from lineaments.

We corrected these lines indicating that the distance between knickpoints and lineaments do not display a clear trend.

21) 192-194: The authors state the faults ‘offset the deformation pattern inferred from marine terraces’ – is this the deformation pattern associated with the large-scale subduction zone deformation? It would be better to be specific here, and clarify how the uplift of terraces due to the PIF has been untangled from the deformation of the coast due to the megathrust.

We rewrote the line, indicating the “offset of the elevation of marine terraces”.

We included the following to Section 3.3:

Marine terraces in the Maule region show two wavelengths of deformation, long wavelength structures (>100 km) associated with deformation of the coast due to the megathrust and short-wavelength structures associated with crustal faults (Jara-Muñoz et al., 2015). The pattern of deformed marine terraces in the PIF area exhibit a short wavelength (~10 km), which is an order of magnitude smaller than the deformation related to the megathrust, therefore the effect of megathrust deformation can be discarded.

22) 194-197: Is there a figure that shows the two models?

We indicate fig. 4D in the text.

23) 219-221: These two observations are in agreement with one another – there is no need to use the word ‘whereas’.

We removed whereas.

24) 365: 68,921 models?

We corrected this typo.

Figures:

25) Figure 1: What magnitude is the aftershock catalogue complete to? It would be helpful to have a sense of what magnitudes are plotted on this figure.

We included the value in the caption.

26) Figure 2: The label ‘edge of seismicity’ is a little confusing because it looks like it is pointing to the two dashed lines (but the seismicity occurs beyond these according to figure 1). This could be clarified as ‘NW boundary of seismicity’. It’s not clear what the heavy dashed black lines represent – these, along with the ‘lineaments’ should be included in the key because the various annotations on this figure are confusing.

We removed the edge of seismicity including now the Pichilemu faults inferred from aftershock seismicity.

27) Figure 4: I suggest using a different colour scales for (a) and (d). The colour scales are similar but different – in that the same colours are used but in a different order, which is strange, and they are showing two different processes (MIS stage vs LOS), and there is no need for them to be the same.

We change the color scale as suggested.

28) Figure 5: It’s not clear what the purple boxes represent in this figure. It would be better to include a 2D or 3D plot of the CFS change.

We now include 2D CFS models of the Maule earthquake, interseismic and three synthetic scenarios to the new Fig. 6. The former figure 5 was moved to Supplementary materials (Fig. S12) including details on the pink and blue lines in the caption.

Supplement

29) Figure S11 – there are some very strong residuals in the S11d – what is being missed in the modelling?

We modified the coseismic elastic dislocation model adjusting the dip based on the recent filtered aftershock seismicity of Calle-Gardella et al. (2021), We also expanded the area of comparison between model and LOS and including slip along F2, which resulted in residuals decreasing to <5 cm near the fault and to <10 cm in the far field (Fig. S8).

30) Figure S12 – which of these scenarios is closed to the Maule earthquake – and what is the resulting pattern of CFS change?

The Maule earthquake was characterized by the rupture of two subsegments. We create synthetic slip scenarios simulating these segments. We explained this in a new figure 6 and in the Section 4.2.

Reviewer #4 (Remarks to the Author):

This is a review of “The cryptic seismic potential of blind faults revealed by off-fault geomorphology, Pichilemu, Chile”. This manuscript presents novel results on blind active faulting on the Pichilemu Fault in the upper plate of the subduction zone, central Chile. The study nicely integrates several different datasets involving field work, remote sensing analysis, and modeling. The study analyzes on- and off-fault geomorphology using morphotectonic and fluvial network analysis, uplifted marine terraces, and luminescence dating, determines the total slip and fault geometry using InSAR, seismicity, and modeling, and addresses reactivation potential of the fault during megathrust earthquakes using Coulomb failure stress modeling. The results indicate that the Pichilemu fault is a blind fault that has a slip rate of 0.42mm/yr and a recurrence interval an order of magnitude higher than the recurrence intervals of megathrust earthquakes on the subduction zone. The findings are broadly applicable and will be of interest to those studying seismic hazards and active faulting in subduction zones (as well as seismic hazards in general), in addition to those studying the tectonics of Chile or the Andes. The conclusions are generally supported by the observations and interpretations presented in the manuscript but do require some further clarification and comparison to previous similar studies.

As I am not an expert in the modeling aspects of this study, my review largely focuses on the tectonic geomorphology methods, results, and interpretations. Below I outline several major issues that should be addressed to help strengthen the message and clarify the methods and interpretations for readers. I also include a list of minor comments by line number, largely to clarify statements in the text.

Major issues include:

A) The interpretation of the lidar that demonstrates the “on-fault” tectonic geomorphology requires additional documentation or explanation to convincingly show or describe possible surface deformation. For example, for the lidar interpretation, there are several other linear features that extend farther than the mapped lineaments, but these are not mapped as lineaments. Aligned linear drainages extend farther northwest from one of the lineaments shown in Fig. S1B. The lineament shown in Fig. S1C is also not convincing, at least as shown. Also note that the white arrows depicting the ends of lineaments in Fig. 2E do not match the extents of the black lines in Fig. 2A and Fig. S1B. The red lines drawn in Fig. 2F do not match with the lineaments shown in Fig. 2A or 2E. It appears that the top 2 profiles (1 and 2) should align with the lineament shown in 3, with another lineament immediately to the NE. Finally, it is not clear what “distance to lineament” on the x-axis represents – perhaps label on Fig. 2E for clarity?

We appreciated and followed these suggestions:

- We included and discussed additional lineaments at the western part of the area (Section 3.1 and 4.3 and Figures 2 and S1).
- We removed the lineament with a weak geomorphic expression in the southeastern part of the area (Fig. 2).
- We included a slope/aspect map to highlight the trace of the lineaments in fig. S1.
- The arrows in figure 2E were corrected. In figure 2A, 2F and S1, the lineaments are now labeled as L3, L4, L5, for clarity.
- We extended the description in the caption of figure 2F, indicating that the profiles are centered on the trace of the lineaments.
- We complemented the descriptions of the lineaments in Section 3.1 including the lithology, slope and red relief maps figure S1.

B) The relationship between the lineaments mapped at the surface, the zone of high seismicity depicted in Fig. 2, and F1 and F2 isn't clearly described. In addition, it is also unclear what “Footwall” and “Hanging wall” in Fig. 2A refer to – are these dashed lines the footwall and hanging wall only for F1? It appears from comparing Fig. 3 with Fig. 2 and Fig S1 that the Fault F1 lies between the dashed hanging wall and footwall lines, but then the zone of high seismicity appears to define F1 and F2 in Fig. 1C. Perhaps these relationships could be clarified in a figure or in the text?

We removed the lines of zone of high seismicity and include now the traces of faults F1 and F2 as inferred from aftershock seismicity. The relationship between F1 and F2 and the lineaments are now described in Section 3.1 and discussed in Section 4.3.

The lineaments are oblique to F1 and F2, and it is unclear how they represent “on-fault” tectonic geomorphology, especially if the fault zone is determined to be a distributed zone that is ~10 km wide.

We tested the hypothesis that these lineaments represent “on-fault” features associated with surface deformation. Based on our observations we discarded the lineaments as fault-scarps generated by earthquakes occurred along PIF. Instead, we observe that catchment metrics display a distributed zone of surface deformation ~10 km wide. We rewrote part of section 3.1 adding this: “The drainages crossing through lineaments L1 and L2 are associated to low K_{sn} values suggesting that if these lineaments are faults, they might be inactive or affected by very low displacement rates or they might respond to differential erosion rather than surface faulting”.

C) The interpretation of the catchment and drainage metrics that demonstrate the “on-fault” tectonic geomorphology requires additional documentation or explanation to convincingly show or describe variable surface deformation.

We included additional catchment metrics (local relief, slope, and catchment asymmetry) and we extended the descriptions and quantitative comparisons between catchments by including the new Fig. S2, and extending descriptions in Section 3.1.

1) The catchment asymmetry and alignment of channels/elongated basins that are subparallel to the dashed lines in Fig. 2 is perhaps more convincing evidence of a zone of blind faulting being expressed at the landscape rather than the oblique lineaments, but these should be described in more detail. The catchment asymmetry would be more convincing if the numbers for the asymmetry factor were presented and discussed.

Following the observations of the reviewer, we included a description of the drainage morphology of catchment C2 and the variability of the symmetry factor of C1 and C2 in figure S2B.

2) This discussion of the differences between the two catchments would be strengthened if more quantitative metrics (instead of qualitatively discussing observations) were discussed to clearly show that one catchment may be uplifted. For example, what about drainage basin relief or relief ratio, drainage basin slope, or some other indices of tectonic activity? Are there differences between the two catchments using some of these metrics?

As previously mentioned, we included additional morphometric parameters in Section 3.1 and Fig. S2. In addition, we expanded the methods Section 5.2.

3) In addition, the easternmost part of C2 appears to be located in the footwall, a part of the landscape presumably uplifting, yet there are no knickpoints or higher channel steepness in that region. How do the authors explain the lack of evidence of uplift in this part of the catchment, if the presence of knickpoints, increased relief, and higher channel steepness is used to explain uplift of C1?

We explain the lack of knickpoints eastwards of C2 because this area is near the termination of fault F1. Thus, this part of C2 is experiencing less surface deformation than the area westwards. This is also shown by decreasing catchment symmetry factors eastwards. To clarify this, we included the faults in Fig. S1 and this observation in Section 3.1.

4) The bedrock lithology and any variations in precipitation over the catchments and their effect on the tectonic geomorphology is also unclear. Early in the manuscript, the bedrock is described as crystalline metamorphic bedrock and assumed to be homogeneous. Later in the manuscript (L308-323), the bedrock is described as more heterogeneous metasedimentary and metavolcanics units with

an overprint of Cenozoic brittle faulting. If the latter description is more accurate, what effect does that have on the catchment and drainage metrics used to infer deformation? For example, are any of the knickpoints or zones of higher channel steepness associated with any of these subtle changes in the bedrock lithology? In Figure 2, there appears to be several prominent NNE-SSW ridges at the southern end of C2 (one of them corresponding to a knickpoint) and just west of C2, in addition to a NE-SW elongate drainage pattern on the SW side of C2. Is this controlled by the bedrock lithology or Cenozoic brittle faulting? Regarding the climate, is there any variation in the precipitation or vegetation over the catchments that may impact how the fluvial network responds?

We extended the description of lithological units, in Sections 2 and 3.1, and including a new map in Fig. S1. We also included the lineaments indicated by the reviewer and describe their relation to lithological contacts in Section 3.1. The description of lineaments and their significance was extended and we included a new figure S1 supporting the analysis. We included rainfall estimates and statistics on catchments C1 and C2 in Figure S2 and Section S1, and describe their influence on catchment metrics in Section 3.1.

5) There appear to be minor inconsistencies in the marine terrace mapping presented in Fig. 3a and the profiles shown in Fig. S4, namely, the mapping and extent of marine terrace MIS5c/e.

The lower marine terrace is a sedimentary package overlying an erosion surface of inferred age MIS 5e and deposited during subsequent sea-level drop after the highstand. We corrected the age attributed to the sedimentary sequence to MIS 5d based on the curves of Bintanja et al. (2005) and Rohling et al. (2009) and corrected the figure S4 including also this information in the caption. This correction did not change our results and interpretations. In addition, we revised the profiles in figure S4 now including Holocene dunes.

We will provide additional supplementary materials including the corrected shoreline angles as kmz files for rapid visualization in Google Earth and also including the swath profiles to facilitate the reproducibility of our estimations.

6) In Fig. 3a, the relatively flat, uplifted area NE of F1 (between “Dune Fields” label and the white star) isn’t mapped as a marine terrace, yet in the profiles in Fig. S4, this area is mapped as MIS5e. Could the authors clarify or revise for consistency?

The part indicated is a field of active dunes covering in part the MIS 5e marine terrace level, for clarity we marked this zone better in Fig. 3 and S4A also including the Holocene deposits in the profiles of Fig. S4B. This part was excluded from the sediment thickness map, the basement discontinuity was inferred for profile 4 in fig. S4B.

7) There have been previous studies (e.g., Farias et al., 2011, Tectonics – cited in this manuscript but not discussed in detail) that discuss the geometry of the Pichilemu fault from seismicity and geomorphology and use Coulomb stress analysis to discuss triggering and activity of the fault. This manuscript would greatly benefit from discussing how their results are similar to or different from previous similar work on the geometry and activity of the Pichilemu fault to improve our understanding of the fault and its associated seismic hazard.

We included a brief comparison between our interpretations on the activity of PIF during the seismic cycle and those from Farias et al. (2011) in the discussion, Section 4.2.

Minor Comments:

8) L104: These lineaments appear to be oriented obliquely to the zone of high seismicity, but the relationship between these lineaments, the zone of high seismicity, and F1 and F2 isn’t clearly described. I see this is briefly addressed in L320-323, but this statement appears to be highly speculative as no detailed information on the bedrock lithology heterogeneity nor how or why these secondary fault tip bending structures are at such oblique angles to F1 and F2.

We included further information on bedrock lithology and metamorphic fabrics, including a geologic map and a detailed analysis of the lineaments in Fig. S1. We described the lineament orientations with respect to faults F1 and F2 in Section 3.1 and discuss the potential factors controlling these orientations in Section 4.3.

9) L117-L120: These statements would be clearer if they referenced specific features in C1 and C2 that demonstrated differences in the hanging wall and footwall. The higher relief and steeper channels in C1, interpreted to be in the uplifted footwall vs the lower relief and lower channel steepness indices in C2 in the hanging wall, for example. As mentioned above, this discussion would be significantly more convincing if quantitative metrics were presented and discussed.

We incorporated a set of catchment metrics to sustain our interpretations, including quantitative estimation of the catchment asymmetry factor and orientation, catchment relief, and slope. These new analyses are included in Figure S2 and discussed in Section 3.1. We also extended their description in the methods, Section 5.2.

10) L119: It is unclear how knickpoints were defined and picked, please add to this section or to the appropriate place in the methods section.

We included the methods used for knickpoint mapping in section 5.2

11) L123: in line 105, it states that there are NW-SE subparallel lineaments associated with slope breaks, but here in line 123, it says no slope breaks were observed. Please clarify.

The lineaments are characterized by very small slope breaks. We clarified this in Section 3.1 to better support our interpretation that they do not represent surface ruptures of past earthquakes.

12) L130: If the distance between the knickpoints and lineaments is uniform, why wouldn't that suggest a relationship to the active faulting?

Thanks for noticing this mistake. We rewrote these lines indicating that the distance between knickpoints and lineaments do not display a clear trend. This suggests that that knickpoints are not related to the potential active fault scarps.

13) L131: The relationship of the knickpoints and the distance to the lineament isn't clear if it's along-channel distance or measured in a straight line?

We changed this line to: "the linear distance between knickpoints and lineaments".

14) L311-313: In this paragraph, the description of the bedrock, previously characterized as homogeneous, appears to say that is the bedrock is heterogeneous, with different sections of metasedimentary and metavolcanics rocks that have an overprint of Cenozoic exhumation. Could the authors clarify the bedrock lithology? Along these lines, a geologic map in the supplementary material could be a useful addition.

We added further details on the bedrock characteristics in Section 2 and 3.1 including a geologic map in figure S1.

15) L365: As I am unfamiliar with this modeling technique, how is it possible to generate a fraction of a model? Perhaps specify or clarify for readers less familiar with this technique?

Indeed, it is not possible to generate a fraction of a model. By mistake we used a point instead of a comma when describing the number of models generated. For clarity, we included all the details of parameters and ranges and the number of models in the new Table S1.

16) L389: The only metrics presented in the manuscript include catchment asymmetry, channel steepness, and chi values – if the authors used other metrics that did not show anything, perhaps that should be clarified here. The asymmetry factor also isn't presented – the observations of catchment asymmetry appears to be qualitative. If the asymmetry factor was calculated, that should be mentioned.

We included additional catchment metrics such as slope, local relief, and catchment asymmetry to Section 3.1 and Figure S2.

17) L398: Is there any difference in precipitation across the catchments, from orographic precipitation, for example? Or is the assumption of uniform climate (and vegetation) an accurate one?

The rainfall is rather similar in both catchments. We provide details and statistics of annual rainfall in Figure S2.

18) Table 1. I suggest listing the total dose rate in Table 1, too.

We added the total dose rates in the Table1.

Reviewers' Comments:

Reviewer #2:

Remarks to the Author:

The authors using SAR data from the Envisat satellite to estimate the surface displacement during the two earthquakes. Before the methods, I would like to see the dates of the interferogram to understand how much postseismic deformation could be in the interferogram? Also, was the interferogram ascending or descending?

I agree that the forward model fits the data, and appreciate the figures that the authors added to the Supporting information to show this.

For what it is worth, I strongly disagree that an inverse model is redundant and adds unnecessary complexity. Inverse models are typically presented and preferred over forward models because they allow for variability in the slip patterns and a better fit to the data. Likely, the forward model presented here would be a fairly "simple" model from a set of inverse models that you produce that fit the data well. The inverse model would support adding some complexity to the slip pattern.

Specifically, for a Mw 6.9 and Mw 7.0 earthquake, it is likely that slip was not constant along the fault plane. An inverse model would support solving for variable slip along the fault plane which could allow for a slip gradient along the shallow portion of the fault. For the shallow slip variations, an inverse model would make for a stronger conclusion than a forward model where the authors test the upper depth of slip.

I agree that the models here fit the data well and are certainly not overly complex, so maybe the forward model is ok. Still, I think the paper could be improved with an inverse model.

Even for a forward model, there is very likely to be a strong trade-off in slip between the two fault planes as they are overlapping and have a similar dip. Could the authors describe any potential trade-off in slip and any implications for the coulomb stress and if not, make a statement that there is not tradeoff.

Reviewer #4:

Remarks to the Author:

This is a re-review of "The cryptic seismic potential of blind faults revealed by off-fault geomorphology, Pichilemu, Chile". The authors have done an excellent job addressing the reviewers' comments on many aspects of the manuscript, including better explaining and documenting the geomorphic evidence of deformation, the forward modelling, and improving the overall presentation of the interpretation and data analysis. The conclusions are now supported by the observations and interpretations presented in the manuscript. Following these revisions, I do not see any major issues with the manuscript; however, there are a few minor comments that would improve the clarity and flow. Importantly, the manuscript does still contain a number of very minor typos and grammatical mistakes – I suggest a careful proofread. In addition, I am still confused by the on- vs. off-fault terminology, which may mean different things to different people. In some sense, even the tectonic geomorphic analysis can be considered "off-fault" because it is documenting deformation over a wide zone. I understand the authors' use of on- vs. off-fault terminology, but wonder if it's the best wording given the general understanding of on- vs. off-fault in the broader community. As I stated in my first review, this is an interesting study that integrates a variety of observations and modeling from different temporal and spatial scales, and will be of interest of the broader community studying seismic hazards and active faulting in subduction zones (Chile/Andes and elsewhere).

Additional minor comments:

L72: Of course this assumes the same coseismic slip distribution in each event, right? I realize there is a detailed discussion of slip per event in section 4.1, but perhaps it would be informative for the reader to state it up front, too?

L81: These two phrases should likely be different sentences, or the comma could be changed to a semi-colon?

L267: I suggest changing this language – can the deformation really be discarded, or perhaps it is better to state that it is negligible or minimal here compared to the shorter wavelength deformation from the crustal faults?

Figure 2: I appreciate the additional/better labeled lineaments included now on this figure. It may be useful to label L3, L4, and L5 on panel (E) as well. I see they are labeled on (A) and (F), but given that the swath profile locations are shown on (E), it would be helpful to the reader to also have the lineaments labeled on the same figure. I also suggest adding a north arrow to (E) as the box is rotated compared to the orientation in (A). In addition, I suggest adding directions to the swath profiles in (F) to better orient the reader, in addition to a y-axis label.

Figure 3: I suggest changing the profile name to something other than X-Y to avoid confusion with the profiles in Figure 1, unless the profile lines are the same (it appears they are in similar orientations?).

Figure S1: This figure really helps examine the lineaments in more detail, and I appreciate both the red-relief and the slope-aspect maps to evaluate the lineaments across the landscape. I suggest increasing the “N” on the north arrow labels on (D) through (G); they are very hard to see and it took me a while to realize they were north arrows and not marking something else!

Figure S4: I suggest labeling directions on the profiles in (B) to better orient the reader.

Figure S12: “Northern” is misspelled in (B) and (D)

REVIEWER COMMENTS

Reviewer #2 (Remarks to the Author):

The authors using SAR data from the Envisat satellite to estimate the surface displacement during the two earthquakes. Before the methods, I would like to see the dates of the interferogram to understand how much postseismic deformation could be in the interferogram? Also, was the interferogram ascending or descending?

These details were included in Section 3.3 also in table S2 (supplementary materials) and in Section 5.1.

I agree that the forward model fits the data, and appreciate the figures that the authors added to the Supporting information to show this.

For what it is worth, I strongly disagree that an inverse model is redundant and adds unnecessary complexity. Inverse models are typically presented and preferred over forward models because they allow for variability in the slip patterns and a better fit to the data. Likely, the forward model presented here would be a fairly "simple" model from a set of inverse models that you produce that fit the data well. The inverse model would support adding some complexity to the slip pattern.

We agree with the reviewer and we included the inverse model in the Supplementary materials (Fig. S11) and discussed its relevance in Section 3.3. We also included inverse modelling information in the methods Section 5.1.

Specifically, for a Mw 6.9 and Mw 7.0 earthquake, it is likely that slip was not constant along the fault plane. An inverse model would support solving for variable slip along the fault plane which could allow for a slip gradient along the shallow portion of the fault. For the shallow slip variations, an inverse model would make for a stronger conclusion than a forward model where the authors test the upper depth of slip.

I agree that the models here fit the data well and are certainly not overly complex, so maybe the forward model is ok. Still, I think the paper could be improved with an inverse model.

We agree and included the inverse model results, which basically show what the reviewer predicted – a relatively simple slip distribution formed by an irregular patch of slip that grades upwards to ~6 km depth. The mean slip of both models agree within confidence bounds and both models suggest a blind fault.

Even for a forward model, there is very likely to be a strong trade-off in slip between the two fault planes as they are overlapping and have a similar dip. Could the authors describe any potential trade-off in slip and any implications for the coulomb stress and if not, make a statement that there is not tradeoff.

We thank the reviewer for pointing out this issue. We evaluated the trade-off by plotting the slip dependency for independent variables and in the NRMSE space. Our analysis suggests a minimal dependency (less than the 10%) between slip of both faults, which does not affect the results of our study. We added an explanation to Section 3.3 and Fig. S12.

Reviewer #4 (Remarks to the Author):

This is a re-review of "The cryptic seismic potential of blind faults revealed by off-fault geomorphology, Pichilemu, Chile". The authors have done an excellent job addressing the reviewers' comments on many aspects of the manuscript, including better explaining and

documenting the geomorphic evidence of deformation, the forward modelling, and improving the overall presentation of the interpretation and data analysis. The conclusions are now supported by the observations and interpretations presented in the manuscript. Following these revisions, I do not see any major issues with the manuscript; however, there are a few minor comments that would improve the clarity and flow.

Importantly, the manuscript does still contain a number of very minor typos and grammatical mistakes – I suggest a careful proofread.

We make an effort to correct all typos and mistakes, in addition, the manuscript was revised a native English speaker colleague.

In addition, I am still confused by the on- vs. off-fault terminology, which may mean different things to different people. In some sense, even the tectonic geomorphic analysis can be considered “off-fault” because it is documenting deformation over a wide zone. I understand the authors’ use of on- vs. off-fault terminology, but wonder if it’s the best wording given the general understanding of on- vs. off-fault in the broader community.

We agree that these terms can lead to confusion, the geomorphic analysis of drainages and lineaments was carried directly on the trace of the fault delineated by the seismicity. Instead, the analysis of marine terraces covered a much bigger area. We included a statement in the introduction to clarify the use of on- and off-fault to distinguish between surface deformation that has occurred along the fault trace and within the zone surrounding the fault, respectively.

As I stated in my first review, this is an interesting study that integrates a variety of observations and modeling from different temporal and spatial scales, and will be of interest of the broader community studying seismic hazards and active faulting in subduction zones (Chile/Andes and elsewhere).

Additional minor comments:

L72: Of course this assumes the same coseismic slip distribution in each event, right? I realize there is a detailed discussion of slip per event in section 4.1, but perhaps it would be informative for the reader to state it up front, too?

We agree and included a statement in line 72 indicating a characteristic slip behavior of the megathrust.

L81: These two phrases should likely be different sentences, or the comma could be changed to a semi-colon?

We changed the comma by a semicolon.

L267: I suggest changing this language – can the deformation really be discarded, or perhaps it is better to state that it is negligible or minimal here compared to the shorter wavelength deformation from the crustal faults?

We changed “discarded” by “negligible compared to the shorter wavelength of crustal faults”.

Figure 2: I appreciate the additional/better labeled lineaments included now on this figure. It may be useful to label L3, L4, and L5 on panel (E) as well. I see they are labeled on (A) and (F), but given that the swath profile locations are shown on (E), it would be helpful to the reader to also have the lineaments labeled on the same figure. I also suggest adding a north arrow to (E) as the box is rotated compared to the orientation in (A). In addition, I suggest adding directions to the swath profiles in (F) to better orient the reader, in addition to a y-axis label.

We included all the suggested modifications to Figure 2.

Figure 3: I suggest changing the profile name to something other than X-Y to avoid confusion with the profiles in Figure 1, unless the profile lines are the same (it appears they are in similar orientations?).

We label the profile P-Q to avoid confusions.

Figure S1: This figure really helps examine the lineaments in more detail, and I appreciate both the red-relief and the slope-aspect maps to evaluate the lineaments across the landscape. I suggest increasing the “N” on the north arrow labels on (D) through (G); they are very hard to see and it took me a while to realize they were north arrows and not marking something else!

We increased the size of north arrows as suggested.

Figure S4: I suggest labeling directions on the profiles in (B) to better orient the reader.

We include the profile orientations.

Figure S12: “Northern” is misspelled in (B) and (D)

We corrected this typo

Reviewers' Comments:

Reviewer #2:

Remarks to the Author:

The authors have addressed my comments. I believe the manuscript is suitable for publication.